# VisionLaw: Inferring Interpretable Intrinsic Dynamics from Visual Observations via Bilevel Optimization

**Jiajing Lin**[1], **Shu Jiang**[2], **Qingyuan Zeng**[2,3], **Zhenzhong Wang**[1], **Min Jiang**[1]*
[1]School of Informatics, Xiamen University.
[2]Institute of Artificial Intelligence, Xiamen University.
[3]The Hong Kong University of Science and Technology (Guangzhou)
`{jiajinglin, shujiang, 36920221153145}@stu.xmu.edu.cn`
`{zhenzhongwang, minjiang}@xmu.edu.cn`

## ABSTRACT

The intrinsic dynamics of an object governs its physical behavior in the real world, playing a critical role in enabling physically plausible interactive simulation with 3D assets. Existing methods have attempted to infer the intrinsic dynamics of objects from visual observations, but generally face two major challenges: one line of work relies on manually defined constitutive priors, making it difficult to align with actual intrinsic dynamics; the other models intrinsic dynamics using neural networks, resulting in limited interpretability and poor generalization. To address these challenges, we propose VisionLaw, a bilevel optimization framework that infers interpretable expressions of intrinsic dynamics from visual observations. At the upper level, we introduce an LLMs-driven decoupled constitutive evolution strategy, where LLMs are prompted to act as physics experts to generate and revise constitutive laws, with a built-in decoupling mechanism that substantially reduces the search complexity of LLMs. At the lower level, we introduce a vision-guided constitutive evaluation mechanism, which utilizes visual simulation to evaluate the consistency between the generated constitutive law and the underlying intrinsic dynamics, thereby guiding the upper-level evolution. Experiments on both synthetic and real-world datasets demonstrate that VisionLaw can effectively infer interpretable intrinsic dynamics from visual observations. It significantly outperforms existing state-of-the-art methods and exhibits strong generalization for interactive simulation in novel scenarios. Our implementation is available at `github.com/JiajingLin/VisionLaw`.

## 1 INTRODUCTION

With the advancement of 4D generation Zhao et al. (2023); Bahmani et al. (2024); Jiang et al. (2024a); Ren et al. (2023), realistic interaction with 3D assets has become increasingly feasible, facilitating broad applications in areas like virtual reality, embodied intelligence, and animation Jiang et al. (2024b); Shi et al. (2023); Lu et al. (2024a); Sun et al. (2024; 2025). Among these advances Xie et al. (2024); Lin et al. (2024b), incorporating physical simulation Stomakhin et al. (2013); Müller et al. (2007) stands out as a particularly prominent method, as it enables the generation of interactive dynamics that closely mirror real-world physical behavior. To ensure simulation realism, it is essential to accurately capture the intrinsic dynamics of objects, including material properties (e.g., stiffness) and constitutive laws Chaves (2013), which describe the response behaviors of materials under applied forces.

Humans can roughly infer the intrinsic dynamics of objects merely by observing their motion, and are even capable of predicting how these objects would interact in new scenarios. A fundamental question arises: *can we enable machines to infer the intrinsic dynamics directly from visual observations, as humans do?* Recent methods Xie et al. (2024); Li et al. (2023) have attempted to bridge

---

*Corresponding author: Min Jiang, minjiang@xmu.edu.cn

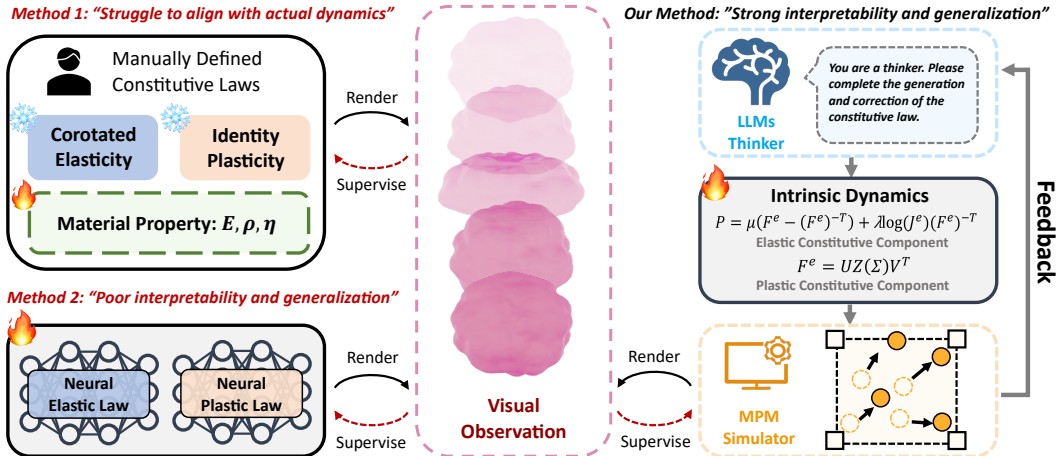

Figure 1: Existing works either rely on manually defined constitutive laws, which struggle to align with actual intrinsic dynamics, or learn neural constitutive laws, which suffer from poor interpretability and generalization. In contrast, our approach can automatically infer interpretable intrinsic dynamics solely from visual observations.

the gap between visual dynamics and physical simulation by incorporating physical simulators (e.g., Material Point Method, MPM Stomakhin et al. (2013)) into 3D representations such as NeRF and 3D Gaussian Splatting (3DGS) Mildenhall et al. (2021); Kerbl et al. (2023). This integration has led to a promising paradigm for inferring the intrinsic dynamics from visual observations. Depending on the type of intrinsic dynamics being inferred, existing methods can be categorized into two groups: material parameter estimation and constitutive law inference.

For material parameter estimation, PAC-NeRF and GIC Li et al. (2023); Cai et al. (2024) estimate material parameters by the supervision of multi-view videos. PhyDreamer, DreamPhysics, and Physics3D Zhang et al. (2024); Huang et al. (2025); Liu et al. (2024) distill visual dynamics priors from video diffusion models to guide the estimation process. However, these approaches typically rely on manually defined constitutive laws, which often fail to align with the complex physical behaviors observed in practice, thereby compromising the accuracy of parameter estimation.

For constitutive law inference, OmniPhysGS Lin et al. (2025) introduces constitutive Gaussians, which assign a suitable constitutive law to each Gaussian kernel from an expert-designed constitutive set. However, such a predefined set often fails to capture the full diversity of real-world physical behaviors. NeuMA Cao et al. (2024) learns neural constitutive laws from visual observations. Despite its effectiveness, it has notable limitations: 1) The learned laws are black-box representations, which lack interpretability and are difficult for humans and LLMs to understand; 2) Due to the lack of physical inductive biases, neural networks tend to mechanically reconstruct visual observations instead of modeling the underlying dynamics, resulting in overfitting and poor generalization.

To overcome the aforementioned challenges, we introduce *VisionLaw*, an interpretable intrinsic dynamics inference framework based on bilevel optimization, which can jointly infer symbolic constitutive laws and their corresponding continuous material properties solely from visual observations. At the upper level, we propose an LLMs-driven decoupled constitutive evolution strategy, which: 1) unleashes the capabilities of LLMs in physical understanding and mathematical reasoning to generate and refine symbolic constitutive hypotheses; 2) introduces a decoupling mechanism to effectively alleviate the search space explosion caused by jointly evolving elastic and plastic components. At the lower level, we construct a vision-guided constitutive evaluation mechanism. Supervised by visual observations, it optimizes the continuous material parameters of a given constitutive law using a differentiable simulator and renderer. The goal is to generate evaluation and feedback that reflect the consistency between the generated laws and ground-truth intrinsic dynamics, which in turn guides the evolution at the upper level. Our contributions are summarized as follows:

- We propose a bilevel optimization framework that unifies constitutive evolution and vision-guided constitutive evaluation, achieving the inference of symbolic constitutive laws and material properties from visual observations.

- We introduce physical inductive biases through LLMs to guide the evolution of constitutive laws. In addition, a decoupled evolution strategy is proposed to markedly improve both search efficiency and solution quality.

Extensive experiments on both synthetic and real-world datasets demonstrate that our method effectively captures the interpretable intrinsic dynamics underlying visual observations and generalizes them to novel scenarios for 4D interaction.

## 2 PRELIMINARIES

### 2.1 CONSTITUTIVE LAWS AND MATERIAL POINT METHOD

In continuum mechanics Chaves (2013), constitutive laws define how materials respond under applied forces. The essential reason why materials like rubber, sand, and water exhibit entirely different physical behaviors lies in the differences in the constitutive laws they follow. To simulate the motion and deformation of materials, we need to solve a system of partial differential equations derived from the conservation of mass and momentum:

$$\frac{D\rho}{Dt} + \rho \nabla \cdot \mathbf{v} = 0, \quad \rho \frac{D\mathbf{v}}{Dt} = \nabla \cdot \mathbf{P} + \rho \mathbf{g}, \tag{1}$$

where $\rho$ denotes the density, $\mathbf{v}$ the velocity field, $\mathbf{g}$ the gravitational acceleration, and $\mathbf{P}$ the stress tensor, which is defined by the constitutive law. The system becomes closed and solvable only after a specific constitutive relation for $\mathbf{P}$ is prescribed.

In this paper, we employ an MPM simulator Stomakhin et al. (2013) to solve the above governing equations because of its versatility in handling various materials. Intuitively, the MPM discretizes the continuum into a set of material points that carry physical quantities such as mass, velocity, and deformation gradient, and updates system states through particle–grid transfers and time integration. At each time step, the particles first project their quantities onto a background grid; the momentum conservation equation is then solved on the grid to compute nodal velocities and accelerations; finally, these updated grid quantities are interpolated back to the particles, advancing their positions $\mathbf{x}$ and deformation gradients $\mathbf{F}$, which describes the local deformation. For more details about MPM, please refer to Appendix D.

Within the MPM framework, two types of constitutive laws must be specified: (1) an elastic constitutive law that describes reversible elastic responses, and (2) a plastic constitutive law that captures irreversible plastic evolution. Their formulations are given as:

$$\varphi_E\left(\mathbf{F}; \theta_E\right) \mapsto \boldsymbol{\tau}, \quad \varphi_P\left(\mathbf{F}; \theta_P\right) \mapsto \mathbf{F}^{\text{corrected}}, \tag{2}$$

where $\varphi_E$ and $\varphi_P$ denote the elastic and plastic constitutive laws, respectively. $\mathbf{F}$ is the deformation gradient, $\boldsymbol{\tau}$ is the Kirchhoff stress tensor, $\mathbf{F}^{\text{corrected}}$ is the corrected deformation gradient after plastic return mapping. The continuous material parameters in the elastic and plastic laws are denoted by $\theta_E$ and $\theta_P$, respectively. Several classical constitutive laws are listed in Appendix E. Despite the availability of many classical constitutive laws, they remain inadequate in capturing the diversity and nonlinear behavior of complex materials. To this end, we propose *VisionLaw*, which infers constitutive laws directly from visual observations.

### 2.2 PHYSICS-INTEGRATED 3D GAUSSIANS

3D Gaussians Splatting (3DGS) Kerbl et al. (2023) represents the scene using a set of anisotropic Gaussian kernels $\mathcal{G} = \{\mathbf{x}_i, \mathbf{A}_i, \alpha_i, \mathcal{C}_i\}_{i \in \mathcal{K}}$, where $\mathbf{x}_i$, $\mathbf{A}_i$, $\alpha_i$, and $\mathcal{C}_i$ represent the center position, covariance matrix, opacity, and spherical harmonic coefficients of the Gaussian kernel $\mathcal{G}_i$, respectively. To render 3D Gaussians into a 2D image from a given view, the color of each pixel can be formulated as:

$$\mathbf{C} = \sum_{i \in \mathcal{N}} \sigma_i \mathbf{SH}(d_i, \mathcal{C}_i) \prod_{j=1}^{i-1} (1 - \sigma_j), \tag{3}$$

where $\mathcal{N}$ denotes a set of sorted Gaussian kernels related to the pixel and view. $\sigma_i$ is the effective opacity, defined as the product of the projected 2D Gaussian weight and opacity $\alpha_i$. $\mathbf{SH}$ computes

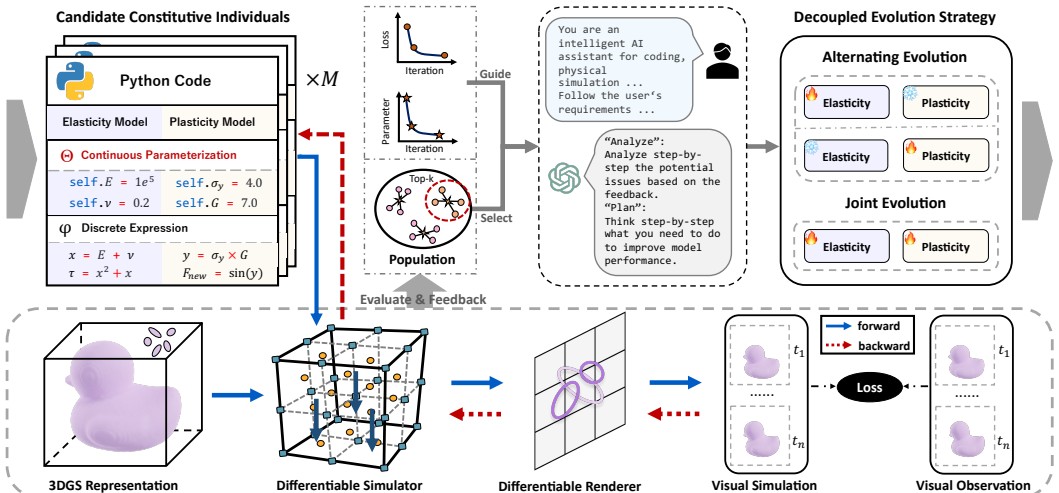

Figure 2: Given a constitutive individual—either predefined at initialization or generated by LLMs—it is embedded into a differentiable MPM simulator for forward simulation. The resulting dynamics are rendered and compared with observations to compute a loss, which is backpropagated to optimize material parameters. This process produces both a fitness score and feedback for the individual. Based on fitness, the top-k individuals are selected and, along with their feedback, encoded into prompts for the LLMs. Guided by the decoupled evolution strategy, the LLMs analyze and refine these constitutive law expressions to generate offspring for the next optimization cycle.

RGB values based on the view direction $d_i$ and spherical harmonic coefficients $\mathcal{C}_i$. Unlike NeRF's implicit form, 3DGS offers an explicit representation that exhibits a Lagrangian nature, facilitating seamless integration with simulation algorithms. Thus, PhyGaussians Xie et al. (2024) pioneers the integration of MPM simulator Stomakhin et al. (2013) into 3DGS, combining physical simulation with visual rendering. Specifically, this method treats Gaussian kernels as particles representing the continuum, and assigns each a time property $t$, material properties $\theta$ (e.g., stiffness). Therefore, given the constitutive law and simulation conditions (e.g., external forces and boundary), MPM can be applied to predict the displacement and deformation of Gaussian kernels at the next time step:

$$\mathbf{x}^{t+1}, \mathbf{F}^{t+1} = \mathbf{\Phi}(\mathcal{G}^t), \tag{4}$$

$$\mathbf{A}^{t+1} = \mathbf{F}^{t+1}\mathbf{A}^t(\mathbf{F}^{t+1})^T. \tag{5}$$

Here, $\mathbf{\Phi}$ is a differentiable MPM simulator, $\mathbf{F}^{t+1}$ denotes the deformation gradient at time step $t+1$ (the subscript $i$ is omitted for simplicity). Gaussian covariance $\mathbf{A}^{t+1}$ can be updated by applying $\mathbf{F}^{t+1}$, which approximates the deformation of the Gaussian kernel. After completing the MPM simulation, a 4DGS representation is constructed, which enables rendering of visual dynamics using Eq. 3.

## 3 METHODOLOGY

In this work, we aim to infer interpretable intrinsic dynamics from a series of visual observations. Formally, given multi-view video observations $V = \{V_1, V_2, ..., V_N\}$ of moving objects along with corresponding camera extrinsic and intrinsic parameters, the goal is to infer the discrete constitutive law expressions and optimize the continuous material parameters in a unified manner. To this end, we propose *VisionLaw*, a novel bilevel optimization framework:

$$\min_{\varphi, \Theta} \mathcal{L}\left(\mathcal{R}\left(\varphi, \Theta, \theta^*; \Phi, \mathcal{G}\right), V\right), \tag{6}$$

$$\text{s.t.} \quad h\left(\varphi, \Theta; \Phi\right) \leq 0, \tag{7}$$

$$\theta^* \in \arg\min_{\theta \in \Theta} \mathcal{L}\left(\mathcal{R}\left(\theta; \varphi, \Phi, \mathcal{G}\right), V\right), \tag{8}$$

where $\mathcal{R}$ is a differentiable renderer defined by Eq. 3. The constitutive law $\varphi$ consists of an elastic law $\varphi_E$ and a plastic law $\varphi_P$. $\Theta$ defines the continuous parameter space for inner-level optimization

$\theta \in \Theta$. $h(\cdot) \leq 0$ refers to the validity of the simulation (e.g. whether a constitutive law $\varphi$ is simulatable). The material parameter $\theta$ includes the elastic parameters $\theta_E$ and the plastic parameters $\theta_P$. For the upper level, based on evaluation and feedback from the lower level, LLMs are employed to generate and refine constitutive law expressions $(\varphi, \Theta)$. At the lower level, given the output $(\varphi, \Theta)$ from the upper level, the optimal continuous material parameters $\theta^*$ are estimated under visual observation supervision, using differentiable rendering and simulation. During this process, evaluation and feedback are provided. The pipeline of the proposed *VisionLaw* is illustrated in Fig. 2.

## 3.1 UPPER-LEVEL CONSTITUTIVE LAWS EVOLUTION

### 3.1.1 LLMS-DRIVEN CONSTITUTIVE EVOLUTION

Recently, LLMs have shown tremendous potential in scientific discovery Yang et al. (2023); Romera-Paredes et al. (2024); Ma et al. (2024); Wang et al. (2026a;b), owing to their strong symbolic reasoning abilities and extensive physical priors. Inspired by this, in the upper-level search, we prompt LLMs to evolve discrete constitutive law expressions. Specifically, we consider LLMs as an intelligent operator and construct an evolutionary search paradigm to iteratively optimize the constitutive law expressions. Each law is represented as a Python code snippet with a clear physical meaning and strong interpretability.

The optimization procedure consists of five stages, which are as follows: i) Initialization: Classical constitutive laws (e.g., purely elastic material models) are introduced as initial individuals. This serves as a physically plausible starting point for the evolutionary process. ii) Fitness Evaluation: Each candidate constitutive individual is passed to the lower level for simulation testing, where its fitness is evaluated from visual observations and the corresponding feedback information is collected. iii) Selection: To improve population diversity and avoid local optima, we first remove duplicate constitutive individuals with fitness differences below a threshold $\epsilon$. Then, we select the top-k constitutive individuals with the highest fitness from the remaining population as "parents" for the next round of evolution. iv) Expression Correction: We prompt LLMs to 1) analyze the parent expression and identify any shortcomings based on its feedback; 2) design an improvement plan and determine how to modify the expression to increase fitness; 3) generate a set of physically plausible constitutive law expressions as candidate individuals. This process is formalized as:

$$\{\varphi^m, \Theta^m\}_{m \in |M|} = \text{LLM}\left(\{\varphi^k, \Theta^k, \mathcal{O}^k\}_{k \in |K|}, \mathcal{P}\right), \tag{9}$$

where $K$ denotes parent size, $M$ denotes offspring size, $\mathcal{O}$ represents the feedback obtained from the lower level and $\mathcal{P}$ denotes the prompt provided to LLMs. v) Iteration: Repeat steps ii) to iv) until a predefined number of iterations is reached. Eventually, the algorithm discovers constitutive laws that not only simulate dynamic behaviors consistent with visual observations but also exhibit strong physical interpretability.

### 3.1.2 DECOUPLED EVOLUTION STRATEGY

In the MPM simulation framework, a complete constitutive law $\varphi$ consists of an elastic part $\varphi_E$ and a plastic part $\varphi_P$, which together govern the system's simulation behavior. However, simultaneous optimization of these components significantly enlarges the search space, increases the difficulty of LLMs search, and hinders convergence to high-quality solutions. To address the above issue, we propose a decoupled evolution strategy that splits the coupled constitutive optimization task into two independently solvable sub-tasks, thereby effectively reducing the search space.

This strategy consists of two phases: 1) Alternating Evolution: In each iteration, we prompt the LLM to optimize only one component of the constitutive law expression (elastic or plastic), while the other remains fixed and is updated in the subsequent iteration. The two components of constitutive laws are optimized alternately across multiple iterations. 2) Joint Evolution: After the alternating optimization phase, we prompt the LLM to jointly optimize both elastic and plastic components to further enhance overall performance. This phase serves as a fine-grained refinement of the existing high-quality expressions from a global perspective. Through the proposed decoupled evolution strategy, we effectively reduce the search space, enhance the stability and efficiency of LLM-based search, and substantially improve the quality of the final constitutive laws.

| Method | BouncyBall | ClayCat | HoneyBottle | JellyDuck | RubberPawn | SandFish | Average |
|--------|-----------|---------|-------------|-----------|------------|----------|---------|
| PAC-NeRF Li et al. (2023) | 516.30 | 15.38 | 2.21 | 137.73 | 15.47 | 1.71 | 114.80 |
| NCLaw Ma et al. (2023) | 56.69 | 2.35 | 0.92 | 11.97 | 3.91 | 1.30 | 12.86 |
| NeuMA Cao et al. (2024) | 1.78 | 1.24 | 1.09 | 10.96 | 1.01 | **1.07** | 2.86 |
| VisionLaw (Ours) | **1.08** | **0.77** | **0.79** | **5.19** | **0.94** | 1.10 | **1.65** |

Table 1: **Quantitative Comparison of Intrinsic Dynamics Consistency on Synthetic Datasets.** The Chamfer distance was employed to quantify the similarity between simulated and ground-truth particle trajectories. Lower values indicate better alignment with ground-truth intrinsic dynamics.

## 3.2 LOWER-LEVEL CONSTITUTIVE LAWS EVALUATION

To effectively evaluate whether a candidate constitutive expression can accurately capture the intrinsic dynamics of motion observed in visual data and to provide high-quality feedback to the upper-level evolution, we design a vision-guided constitutive evaluation mechanism. First, a static 3DGS representation is reconstructed from the first frame of multi-view video inputs. Then, the candidate constitutive law expression $\varphi(\theta)$, with continuous material parameters, is seamlessly embedded into a differentiable MPM simulator. We integrate the MPM simulator with 3DGS to drive the simulation and render the predicted visual dynamics $V$ from given views. The supervised loss between the predicted and observed visual dynamics can be formulated as:

$$\mathcal{L} = \frac{1}{N} \sum_{n=1}^{N} [\lambda \mathcal{L}_2(\hat{V}_n, V_n) + (1 - \lambda)\mathcal{L}_{\text{D-SSIM}}(\hat{V}_n, V_n)], \tag{10}$$

where $\hat{V}_n$ denotes the rendered video from the $n$-th viewpoint, and $\mathcal{L}_2$ is the L2 norm loss. Since both the renderer $\mathcal{R}$ and the MPM simulator $\Phi$ are differentiable, the evaluation loss can be back-propagated to optimize the continuous material parameters $\theta$ as described in Eq. 8. During this process, we collect the loss curve and the material parameter update trajectory as feedback $\mathcal{O}$ to construct the LLMs' prompts. Meanwhile, the minimum loss achieved during optimization is used as the fitness score of the constitutive candidate to guide the selection process at the upper level.

## 4 EXPERIMENTS

### 4.1 EXPERIMENTAL SETUP

#### 4.1.1 IMPLEMENTATION DETAILS

Given multi-view videos of a scene, we follow NeuMA Cao et al. (2024) to perform 3D reconstruction and Particle-GS binding using multi-view images from the initial time step. We use only single-view videos as ground-truth observations to infer intrinsic dynamics across all experiments. For all scenarios, the initial constitutive individual is only defined as a purely elastic model that combines linear isotropic elasticity with identity plasticity. For the upper-level evolution, we employ `GPT-4.1-mini` to generate constitutive hypotheses. Details of the prompt design are provided in Appendix G. The decoupled evolution strategy is executed through four iterations of alternating optimization, followed by three iterations of joint optimization. For lower-level optimization, we conduct MPM simulation Xie et al. (2024) under gravitational acceleration ($9.8m/s^2$). We employ the Adam optimizer with a learning rate of $1 \times 10^{-3}$ to tune the material parameters. For each scene, we perform five independent runs using different random seeds. All experiments are conducted on an NVIDIA A40 (48 GB) GPU. Detailed experimental settings are provided in Appendix A.1.

#### 4.1.2 BASELINES

We compare our method with state-of-the-art intrinsic dynamics inference methods: PAC-NeRF Li et al. (2023), NCLaw Ma et al. (2023), NeuMA Cao et al. (2024), and Spring-Gaus Zhong et al. (2024). PAC-NeRF is capable of inverting material parameters from video input. NCLaw only fits neural constitutive laws to known dynamics, whereas NeuMA extends this by introducing visual information for adaptation. NeuMA is the most relevant work to ours, as it learns neural constitutive laws directly from visual observations. Spring-Gaus models elastic objects by integrating spring-mass system with Gaussian kernels. It achieves intrinsic dynamics inference through the estimation of spring stiffness. All baseline experimental settings follow the original setup.

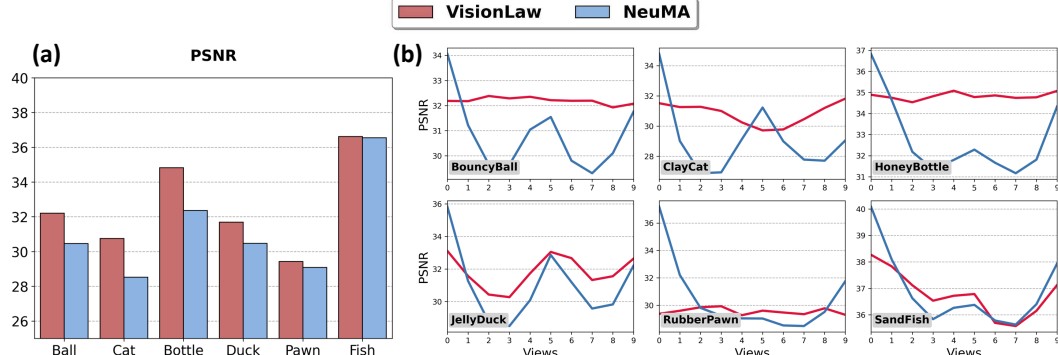

Figure 3: **Quantitative Comparison of Visual Fidelity on Synthetic Datasets.** (a) Average PSNR over all non-training views. Higher PSNR values reflect improved visual fidelity; (b) PSNR comparison at different views, **with View 0 denoting the training view.**

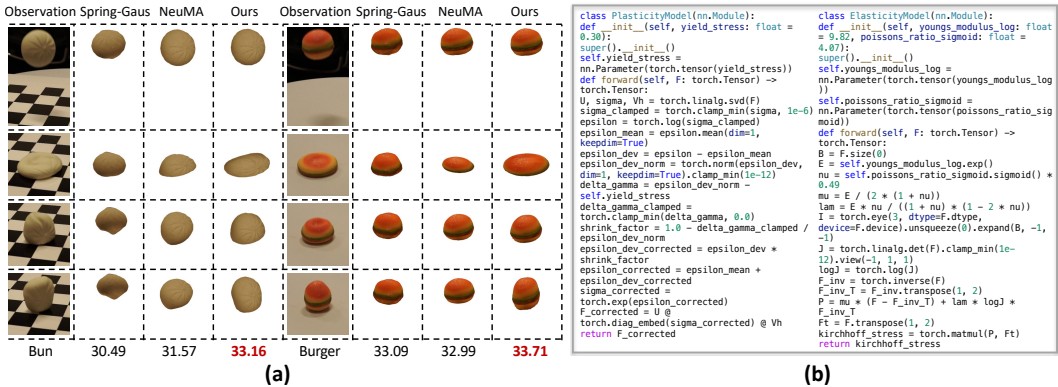

Figure 4: **Comparison on Real-World Datasets.** (a) Quantitative metrics (i.e., PSNR) between the predicted and observed frames are reported in the bottom row; (b) The intrinsic dynamics inferred from the Bun scene, represented as Python code, exhibit strong interpretability.

### 4.1.3 DATASETS AND METRICS

To thoroughly evaluate the effectiveness of our method, we conduct experiments on both synthetic and real-world datasets. For synthetic data, we adopt six dynamic scenes from NeuMA Cao et al. (2024), each with varying initial conditions (including object shapes, velocities, and positions), intrinsic dynamics, and simulation time intervals. Each synthetic scene includes 10 videos captured from different views, each containing 400 frames, and the dataset further provides ground-truth particle trajectories. For real-world evaluation, we conduct experiments on two scenes ('Bun' and 'Burger') provided by Spring-Gaus Zhong et al. (2024). Each real-world scene includes 3 videos captured from different views, each containing 19 frames. More details of the datasets are provided in Appendix A.2. Following prior works Guan et al. (2022); Cao et al. (2024), we use the L2-Chamfer distance Erler et al. (2020) between the simulated and ground-truth particle trajectories to quantify the accuracy of intrinsic dynamics inference. To assess the visual fidelity, we follow 3DGS Kerbl et al. (2023) and employ PSNR, SSIM, and LPIPS as quantitative metrics.

### 4.2 PERFORMANCE ON INTRINSIC DYNAMICS INFERENCE

### 4.2.1 SYNTHETIC DATASET.

**Comparison of Intrinsic Dynamics Consistency.** In synthetic datasets, ground-truth particle trajectories are generated from ground-truth intrinsic dynamics. We evaluate alignment between inferred and ground-truth intrinsic dynamics by measuring the Chamfer distance between simulated and ground-truth trajectories, as summarized in Tab. 1. PAC-NeRF relies heavily on manually designed constitutive laws and is highly sensitive to material parameter initialization. This restricts

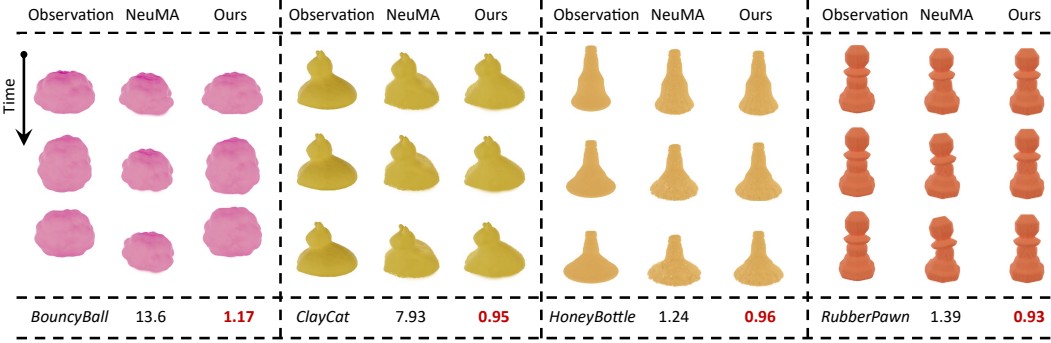

Figure 5: **Generalization to Unseen Observations.** We infer the intrinsic dynamics using only the first 200 frames of visual observation and simulate the subsequent 200 frames. Quantitative metrics (i.e., Chamfer distance) are reported in the bottom row.

its ability to capture actual dynamics, leading to poor performance, especially in complex scenarios such as BouncyBall and JellyDuck. Similarly, NCLaw learns predefined constitutive laws and suffers from the same limitations as PAC-NeRF. NeuMA improves flexibility by learning neural constitutive laws from visual inputs. However, its black-box nature limits interpretability and often leads to overfitting. In contrast, our *VisionLaw* approach achieves the best overall performance across all six benchmarks, with an average Chamfer distance of 1.65, significantly outperforming the baselines. These results demonstrate the superior ability of *VisionLaw* to recover intrinsic dynamics directly from visual observations, while maintaining interpretability.

**Comparison of Visual Fidelity.** To further evaluate visual fidelity, we compute the PSNR between rendered dynamics and ground-truth observations. As shown in Fig. 3 (a), we report the averaged PSNR over all non-training views. The results show that *VisionLaw* significantly outperforms NeuMA, achieving superior visual fidelity. In Fig. 3 (b), we further compare PSNR across different views, including the training view (View 0). NeuMA exhibits pronounced variability, with higher PSNR at the training view and its neighbors (View 1 and View 9), but considerably worse performance on unseen views. This shows that NeuMA tends to overfit the training views, which limits its ability to generalize. In contrast, VisionLaw performs consistently across different views and still produces robust results on unseen views, even when trained on only one. This stability arises from introducing physical inductive biases through LLMs into the evolution of constitutive laws, which effectively mitigates the overfitting commonly observed in purely neural methods. Overall, these findings confirm that our approach not only captures more faithful intrinsic dynamics but also delivers dynamic reconstructions of higher visual fidelity.

### 4.2.2 REAL-WORLD DATASET.

We evaluated our method on a real-world dataset against Spring-Gaus Zhong et al. (2024) and NeuMA Cao et al. (2024), with visual results and PSNR metrics shown in Fig. 4(a). Spring-Gaus models elastic deformation using a spring–mass system, which works well for simple linear behaviors, but fails to capture the complex nonlinear elasticity of real deformable objects. Consequently, its predictions deviate markedly from the ground-truth observations. NeuMA employs neural networks to approximate nonlinear dynamics and capture diverse material behaviors. However, it is sensitive to observation noise and lacks explicit physical constraints, which limits its ability to reproduce the subtle deformations of real-world objects. In contrast, *VisionLaw* integrates a broad range of physical priors through LLMs, providing a strong inductive bias toward physically plausible dynamics. This improves both generalization and learning stability. As shown in Fig. 4 (a), *VisionLaw* generates results that are more consistent with real observations, both visually and quantitatively. These results demonstrate that *VisionLaw* can accurately capture the intrinsic dynamics of deformable objects and highlight its practical effectiveness in real-world scenarios. Meanwhile, Fig. 4 (b) illustrates the inferred intrinsic dynamics in the Bun scenario, expressed in the form of Python code. This form offers strong interpretability, allowing humans to intuitively grasp the physical meaning underlying the formulas, thereby facilitating scientific discovery. Moreover, symbolic expressions provide an implicit regularization effect, which helps prevent overfitting.

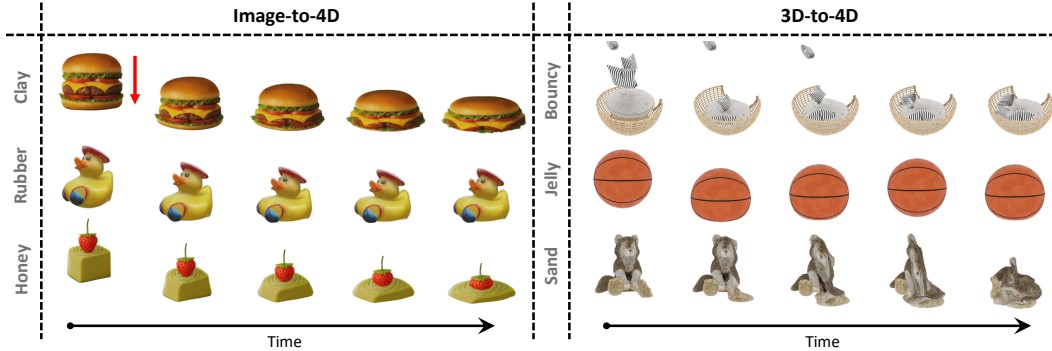

Figure 6: **Generalization to Novel Scenarios for 4D Interaction.** The left text indicates the intrinsic dynamics applied, which are learned from visual observations through *VisionLaw*.

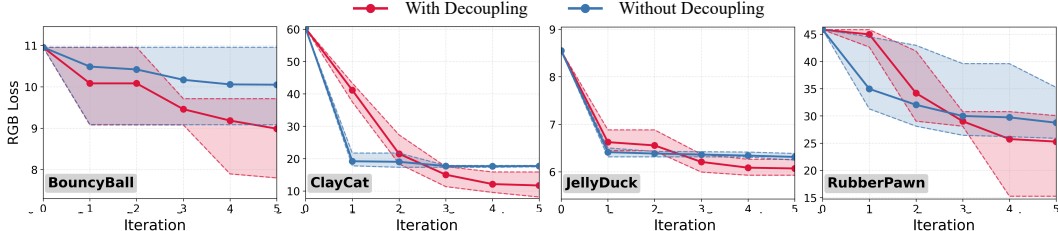

Figure 7: **Ablation Study on Decoupled Evolution Strategy**. The figure shows the loss of the best solution averaged across seeds at different iterations. The shaded area indicates the range between the minimum and maximum values.

## 4.3 GENERALIZATION ANALYSIS AND ABLATION STUDIES

### 4.3.1 GENERALIZATION TO UNSEEN OBSERVATIONS.

We conducted a generalization analysis on four examples, comparing our method with NeuMA Cao et al. (2024). For each scene, the first 200 frames of visual observations were used to infer the intrinsic dynamics, which were then used to predict the next 200 frames. As shown in Fig. 5, NeuMA struggles to generalize beyond the observed frames. Its predictions diverge significantly from the ground truth, likely due to overfitting. In contrast, *VisionLaw* achieves consistently high predictive accuracy across both visual appearance and Chamfer distance metrics, even with limited observation data. We attribute this advantage to the physical inductive bias introduced by knowledge-rich LLMs, which not only improves physical plausibility but also constrains the solution space in a meaningful way. These results highlight that *VisionLaw* combines strong generalization with interpretability, making it practical for forward simulation in previously unseen temporal regimes.

### 4.3.2 GENERALIZATION TO NOVEL SCENARIOS

To further verify the generalization and transferability of the interpretable intrinsic dynamics learned by VisionLaw from visual observations, we apply the dynamics learned from different scenarios to novel 4D generation tasks. The 3D-to-4D and image-to-4D tasks follow the paradigms of Phys-Gaussian Xie et al. (2024) and Phy124 Lin et al. (2024a), respectively, and all experiments are conducted under gravitational conditions. As shown in Fig. 6, all examples generate dynamics consistent with the original observations, such as the slow deformation of clay, the elastic recovery of rubber, and the dispersive behavior of sand. These results demonstrate that the intrinsic dynamics inferred by *VisionLaw* are not only interpretable but also transferable to unseen scenarios, enabling the 4D interaction aligned with real physical behaviors. This cross-scenario generalization opens new possibilities for physics-driven 4D interaction.

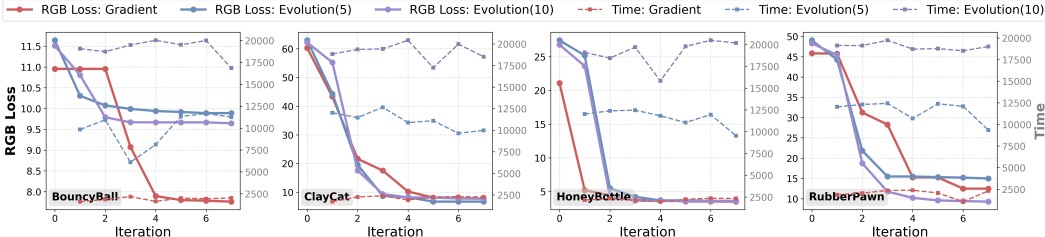

Figure 8: **Comparison across Different Lower-level Optimization Paradigms.** We compare two types of lower-level parameter optimization: gradient-based optimization (default) and evolutionary search, where Evolution($\cdot$) denotes an evolutionary strategy with the specified population size. The left axis report the RGB loss computed against observations, while the right column records the runtime.

### 4.3.3 ABLATION STUDY ON DECOUPLED EVOLUTION STRATEGY

To evaluate the effectiveness of our proposed decoupled evolution strategy, we perform an ablation study comparing two settings over five iterations: 1) with decoupling: the elastic and plastic components are optimized alternately for four iterations, followed by a final joint refinement step; 2) without decoupling: all five iterations are performed with joint optimization. As illustrated in Fig. 7, the decoupled strategy consistently yields lower RGB losses across diverse scenes, indicating it leads to better constitutive law discovery. By decomposing the search into simpler sub-tasks, it narrows the search space, making optimization more efficient. Moreover, the shaded regions are noticeably larger under the decoupled setting, indicating greater solution diversity. This helps avoid early convergence to poor local minima. Overall, the decoupled evolution strategy more effectively unleashes the potential of LLMs by not only sharpening exploitation but also broadening exploration.

### 4.3.4 POTENTIAL ANALYSIS IN NON-DIFFERENTIABLE SIMULATION ENVIRONMENTS

To demonstrate the potential of *VisionLaw* in non-differentiable simulation environments, we compare two types of lower-level parameter optimization: (i) the gradient-based optimization adopted as the default in this work, and (ii) the gradient-free evolutionary strategies implemented via the differential evolution algorithm Storn & Price (1997). For evolutionary search, we evaluate population sizes of 5 and 10, with 10 lower-level optimization iterations—matching the gradient-based baseline. As shown in Fig. 8, even when the lower-level optimizer is replaced with evolutionary search, *VisionLaw* consistently converges to solutions comparable to those from gradient-based optimization. In certain scenarios (e.g., ClayCat and RubberPawn), evolutionary search even achieves superior results. On the other hand, evolutionary search incurs a substantial computational overhead, as each iteration requires multiple forward simulations to evaluate all candidate parameter individuals in the population. Nevertheless, *VisionLaw*'s compatibility with gradient-free evolutionary strategies allows it to function effectively in non-differentiable simulation environments. This highlights its broad applicability and practical significance in the real world.

## 5 CONCLUSION

In this paper, we propose *VisionLaw*, a bilevel optimization framework that infers interpretable intrinsic dynamics directly from visual observations by jointly optimizing the symbolic constitutive law and its material parameters. At the upper level, knowledgeable LLMs are prompted to generate and refine symbolic constitutive laws, thereby introducing physical inductive biases into constitutive evolution. Meanwhile, a decoupled evolution strategy is introduced to reduce the complexity of jointly searching and to improve the solution quality. At the lower level, material parameters are optimized under visual supervision, while evaluation and feedback on intrinsic dynamics consistency are provided to guide the upper-level evolution. This closed-loop design effectively bridges the gap between visual data and physical nature, achieving a balance between interpretability, physical plausibility, and generalization. Experimental results show that our method accurately captures intrinsic dynamics from visual observations and generalizes well to novel scenarios for 4D interaction.

## ACKNOWLEDGMENTS

This work was supported in part by the National Natural Science Foundation of China under Grant No. 52535009. This work was also supported by the National Natural Science Foundation of China under Grant No. 62276222.

## ETHICS STATEMENT

This research adheres to the ethical guidelines outlined by ICLR. We confirm that no human subjects were involved in this study, and all datasets used have been properly sourced and are publicly available. Our methods have been designed with fairness and transparency in mind, ensuring no biases are introduced in the analysis. Privacy and security of data have been prioritized throughout the research, and we comply with all applicable legal regulations. No conflicts of interest or sponsorships have influenced the research outcomes. We are committed to upholding research integrity and have followed appropriate ethical practices throughout the study.

## REPRODUCIBILITY STATEMENT

We have made efforts to ensure the reproducibility of our work. The source code for the algorithms presented in this paper is publicly available at `github.com/JiajingLin/VisionLaw`. Additionally, a detailed description of the experimental setup and datasets is provided in the Appendix. We encourage reviewers and readers to refer to these materials for complete reproducibility.

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

APPENDIX

In this appendix, we will provide: i) more experimental details; ii) more experimental results; iii) related work; iv) implementation details of the MPM algorithm; v) a summary of classical constitutive laws; vi) details of the prompt design. vii) visualizations of the inferred constitutive laws.

## THE USE OF LARGE LANGUAGE MODELS (LLMS)

Large language models (LLMs) were utilized in this work to improve the fluency and clarity of the manuscript. Their application was specifically focused on detailed proofreading to correct spelling errors and ensure grammatical accuracy, as well as refining sentence structures to enhance the readability and logical flow of the paper. It is crucial to note that all scientific contributions, including the core concepts, experimental design, data analysis, and conclusions, were entirely conceived and written by the authors. The LLMs were employed solely as a writing assistance tool and did not contribute to the conceptualization or analysis of the study.

## A   MORE EXPERIMENTAL DETAILS

### A.1   IMPLEMENTATION DETAILS

Given multi-view videos of a scene, we first perform 3DGS reconstruction Kerbl et al. (2023) using the multi-view images from the initial time step. Following NeuMA Cao et al. (2024), we establish relationships between simulation particles and Gaussian kernels via the Particle-GS mechanism. To infer intrinsic dynamics from visual observations, we utilize only single-view videos as ground-truth observations across all datasets. For the upper-level evolution, we employ `GPT-4.1-mini` to generate constitutive hypotheses. **For all scenarios, the initial constitutive individual is only defined as a purely elastic model that combines linear isotropic elasticity with identity plasticity.** The alternating evolution phase consists of 4 iterations. In each iteration, the top 3 individuals are selected, and each generates 6 offspring independently. In the subsequent joint evolution phase, we conduct 3 iterations. In each iteration, the top five individuals are selected to jointly prompt GPT, generating 18 offspring in one shot. For lower-level optimization, we conduct MPM simulations under standard gravitational acceleration ($9.8 \, m/s^2$) within a unit cube domain $[0, 1]^3$. The simulation resolution is set to $32^3$ for synthetic data and $70^3$ for real-world data. We employ the Adam optimizer with a learning rate of $1 \times 10^{-3}$, and perform 10 iterations to tune the material parameters of a single constitutive law. For each scene, we conduct five independent runs using different random seeds: 0, 1, 2, 3, and 4. All experiments are conducted on NVIDIA A40 (48GB) GPU.

### A.2   DATASET DETAILS

The synthetic dataset is derived from NeuMA Cao et al. (2024) and consists of six scenes ('BouncyBall', 'JellyDuck', 'RubberPawn', 'ClayCat', 'HoneyBottle', and 'SandFish'). Each scene records the motion of a single object, providing observations from 10 viewpoints with a total of 400 frames per dynamic sequence. To reduce computational resources, for the synthetic data, we select one frame every five frames from the video to create the training set. This dataset features a variety of material types, ranging from elastic bodies to granular materials, exhibiting diverse dynamic behaviors and complex geometric shapes. Meanwhile, the synthetic dataset also provides ground-truth particle trajectories, which can be used to evaluate the consistency between the inferred and ground-truth intrinsic dynamics. The real-world dataset is taken from Spring-Gaus Zhong et al. (2024) and contains two scenes ('Bun' and 'Burger'). It provides observations from 3 viewpoints, with each dynamic sequence consisting of 19 frames. In all experiments, the initial velocity $v_0$ follows the configuration provided in NeuMA's dataset description. We use only a single frontal view of the object as visual observation to infer its intrinsic dynamics.

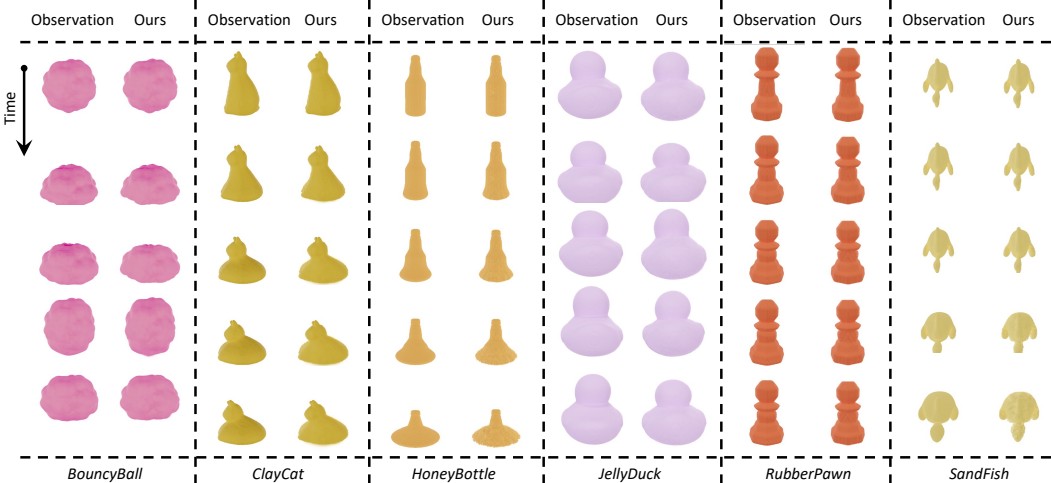

Figure 9: **Visual Results on Synthetic Dataset.** We select the rendered images at frames 1, 100, 200, 300, and 400. *VisionLaw* exhibits dynamics similar to those observed in visual observations.

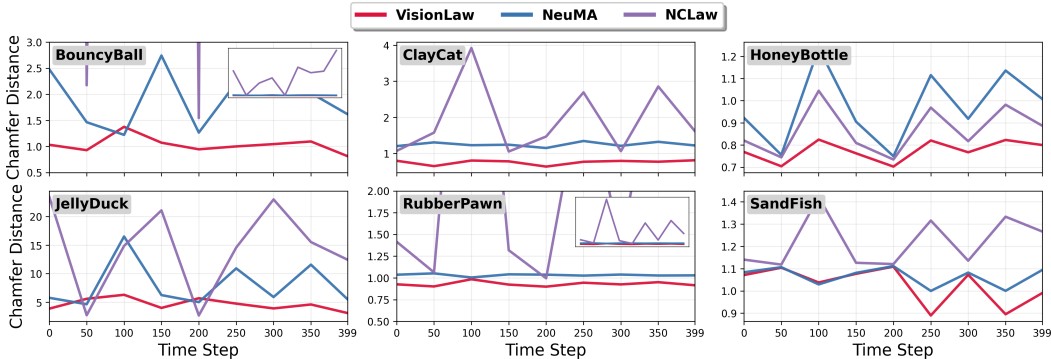

Figure 10: **Comparison of Chamfer Distance at Different Time Steps on Synthetic Dataset.**

# B   MORE EXPERIMENTAL RESULTS

## B.1   QUALITATIVE VISUALIZATION RESULTS

We provide qualitative results on six synthetic scenes to assess the visual fidelity of our method. As shown in Fig 9, we compare rendered outputs from our model with ground-truth observations at selected time frames (1, 100, 200, 300, and 400). Our method accurately reproduces object dynamics over time, showing close alignment with the ground truth across all scenes. These results demonstrate that *VisionLaw* effectively captures complex deformation behaviors with visual realism.

## B.2   QUANTITATIVE COMPARISON OF CHAMFER DISTANCE

As shown in Fig. 10, we compare the Chamfer distance of *VisionLaw*, NeuMA Cao et al. (2024), and NCLaw Ma et al. (2023) across different time steps on the synthetic dataset. NCLaw consistently shows the worst performance. This is because NCLaw can only fit the known dynamics, but fails to adapt to the underlying intrinsic dynamics behind the visual observations. As a result, its error remains high across all objects. NeuMA introduces additional neural network components to capture the mapping between visual observations and intrinsic dynamics. However, due to the lack of physical inductive bias, NeuMA is mainly based on memorization, leading to overfitting and unstable predictions. In contrast, *VisionLaw* distills physical priors from LLMs to refine constitutive laws, thereby incorporating a form of physical inductive bias. This mechanism enhances its abil-

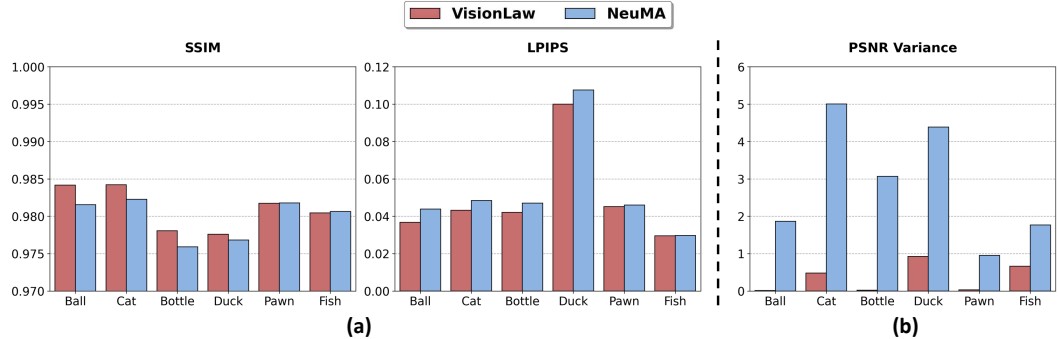

Figure 11: **Quantitative Comparison of Visual Fidelity on Synthetic Datasets.** (a) Average SSIM and LPIPS over all non-training views. Higher SSIM and lower LPIPS values reflect improved visual fidelity; (b) PSNR variance over all views, including training views.

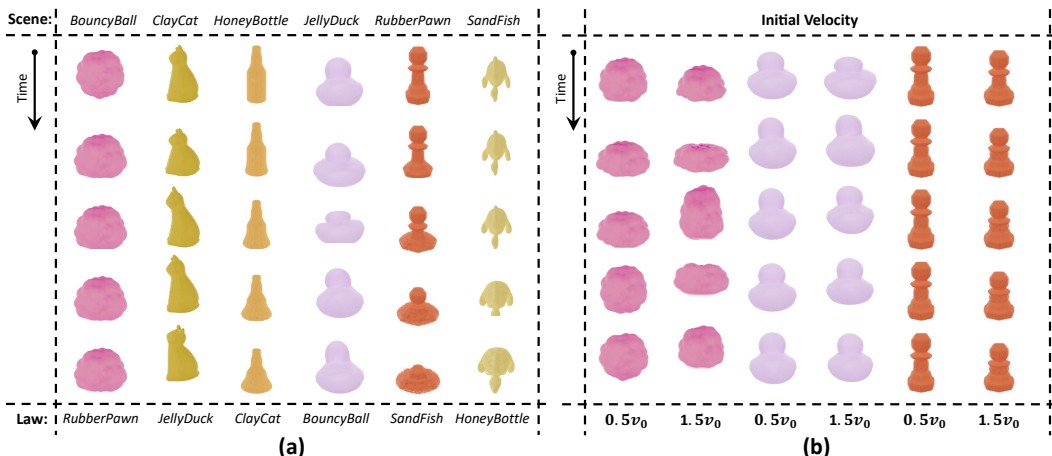

Figure 12: **Generalization Analysis.** (a) Generalization to new scenarios. The top row shows the simulated scenes, while the bottom row presents the intrinsic dynamics inferred for the given scenarios. (b) Generalization to different initial velocities. The bottom row represents the configured initial velocity, expressed as a multiple of the original initial velocity.

ity to discover hidden dynamics, leading to consistently better performance across different objects and time steps. As shown in Fig. 10, *VisionLaw* achieves lower Chamfer distance, demonstrating stronger adaptability to complex dynamics.

### B.3 QUANTITATIVE COMPARISON OF VISUAL FIDELITY

To more comprehensively evaluate visual fidelity, we report average SSIM and LPIPS across all non-training views in Fig. 11 (a). The results show that *VisionLaw* outperforms NeuMA Cao et al. (2024). This confirms that our method not only captures more faithful intrinsic dynamics but also produces dynamic reconstructions with higher perceptual fidelity. We further compute the PSNR variance over all views in Fig. 11 (b), which reflects the generalization to unseen views. NeuMA exhibits high PSNR variance, indicating a tendency to overfit. In contrast, VisionLaw achieves a much lower variance. This demonstrates that, even when trained from a single fixed viewpoint, our method generalizes effectively to novel views by leveraging the physical inductive bias introduced through LLMs.

### B.4 GENERALIZATION ANALYSIS

We first evaluate cross-scene generalization by applying the intrinsic dynamics inferred from one scenario to simulate another. As shown in Fig 12(a), the top row presents the target scenes, while the bottom row shows the intrinsic dynamics inferred from different sources. Despite the mismatch

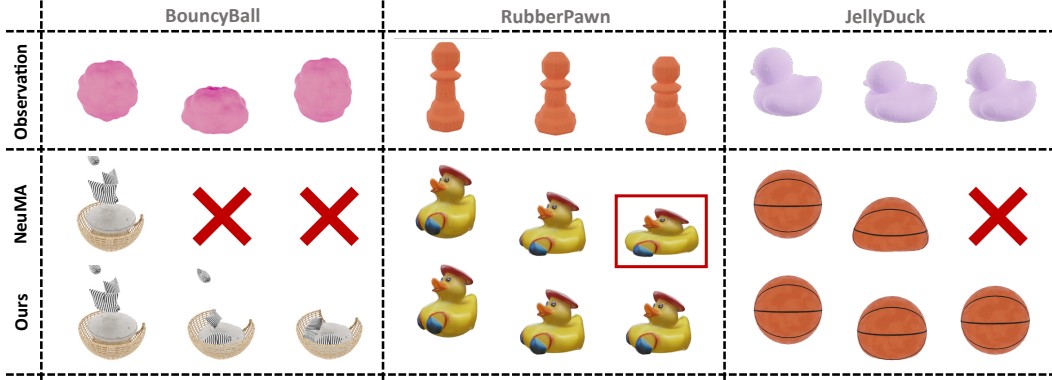

Figure 13: **Comparison of Generalization to Novel Scenarios.** The first row shows the visual observations used to learn the intrinsic dynamics. The rendered frames in each column correspond to the same simulation time step across all methods. Red crosses indicate numerical divergence during simulation, which terminates the rollout.

| Methods | Metric | BouncyBall | ClayCat | HoneyBottle | JellyDuck | RubberPawn | SandFish | Average |
|---|---|---|---|---|---|---|---|---|
| w/o decoupling | Accuracy | 9.08 | 17.31 | 3.64 | 6.24 | 25.88 | 2.24 | 10.73 |
| | Time | 11790 | 10997 | 10891 | **9333** | 10605 | **14439** | **11343** |
| w/ decoupling | Accuracy | **7.80** | **8.13** | **3.54** | **5.93** | **15.25** | **2.17** | **7.14** |
| | Time | **9319** | **10636** | **10699** | 11249 | 10839 | 15689 | 11405 |

Table 2: **Search Efficiency Analysis of Decoupled Evolution Strategy.** Comparison between naive joint optimization (w/o decoupling) and decoupled evolution strategy (w/ decoupling). Accuracy denotes RGB loss (lower is better); Time indicates time consumption in seconds (lower is better).

between the source scene and the target, our method consistently produces physically plausible behaviors. This indicates that the constitutive laws discovered by *VisionLaw* are not merely scene-specific fits but encode transferable physical priors, demonstrating strong cross-scene generalization. We further design experiments under different initial conditions by varying the initial velocity of objects (with the baseline $v_0$ specified in the NeuMA Cao et al. (2024) dataset description). As shown in Fig. 12 (b), the results show that, even with varying initial velocities, the intrinsic dynamics inferred by *VisionLaw* still accurately reflect the object's behavior. This result underscores the robustness of our method in the face of variations in initial conditions, confirming that *VisionLaw* identifies fundamental physical laws that extend beyond the specific configurations used in training.

### B.5 GENERALIZATION COMPARISON

To further validate the generalization capability of *VisionLaw*, we compare it with NeuMA on a 4D interaction generation task, with all experiments conducted under gravity only. As shown in Fig. 13, the intrinsic dynamics learned by NeuMA exhibit clear limitations when transferred to novel scenarios. First, NeuMA frequently suffers from numerical instabilities, causing the simulation to diverge and terminate prematurely, as observed in the BouncyBall and JellyDuck cases. Even when rollouts remain stable, NeuMA produces dynamics that deviate markedly from the reference observations. For instance, the duck undergoes excessive collapse, inconsistent with the visual behavior observed in the RubberPawn reference. These failures stem from NeuMA's lack of physical inductive biases, which leads it to mechanically reconstruct appearance rather than truly capturing the latent intrinsic dynamics—ultimately resulting in poor generalization. In contrast, *VisionLaw* consistently generates stable 4D interactive dynamics that align with the original observations. By incorporating physical inductive biases through LLMs, *VisionLaw* effectively captures the underlying intrinsic dynamics, enabling robust and reliable generalization to unseen scenarios.

### B.6 SEARCH EFFICIENCY ANALYSIS OF DECOUPLED EVOLUTION STRATEGY

To assess the search efficiency of our decoupled evolution strategy, we conducted an ablation study under the same experimental settings described in Sec. 4.3.3. As shown in Tab. 2, the decoupled strategy significantly improves accuracy, reducing the average RGB loss from 10.73 to 7.14, while maintaining comparable computational cost (11343s vs. 11405s). This result demonstrates that the decoupled strategy effectively unleashes the LLM's capability for constitutive discovery by making the optimization more focused, producing superior solution quality without additional computational overhead.

## C RELATED WORK

### C.1 PHYSICS-BASED 4D INTERACTION

Advances in 3D representation methods Mildenhall et al. (2021); Müller et al. (2022); Kerbl et al. (2023); Zeng et al. (2025) (e.g., NeRF and 3DGS) have greatly facilitated the creation of 3D assets Poole et al. (2023); Tang et al. (2024b;a), consequently drawing significant attention to the pursuit of realistic interaction with these assets. To enable physically plausible 4D interaction, recent works have attempted to incorporate various physical simulators Stomakhin et al. (2013); Macklin et al. (2016) with 3D representation. PIE-NeRF Feng et al. (2024) enables meshless nonlinear elastodynamic simulation directly in NeRF via augmented Poisson disk sampling and quadratic generalized moving least squares (Q-GMLS) Martin et al. (2010). Inspired by the Lagrangian nature of 3DGS, PhysGaussian Xie et al. (2024) pioneered the integration of MPM simulator into 3DGS. Phys4DGen Lin et al. (2024b) effectively perceives multiple materials within a single object and automatically assigns material properties by distilling physical priors from MLLMs Achiam et al. (2023), enabling more accurate and user-friendly interactive dynamic generation. VR-GS Jiang et al. (2024b) conducts tessellation via TetGen Hang (2015) to convert 3DGS representations into tetrahedral meshes, enabling fast XPBD simulation and physically plausible interaction in VR.

### C.2 INTRINSIC DYNAMICS LEARNING

Understanding the intrinsic dynamics underlying observational data is highly valuable for interactive simulation Müller & Gross (2004) and scientific discovery Wang et al. (2023). Deep learning Gong et al. (2026; 2024); Zeng et al. (2024); Lu et al. (2024b; 2023) has advanced rapidly and is increasingly being applied to physical simulation Sanchez-Gonzalez et al. (2020); Yang et al. (2025); Wang et al. (2025), with some methods Pfaff et al. (2020); Ummenhofer et al. (2019) using end-to-end networks to model physical laws. However, purely neural approaches often lack physical consistency. NCLaw Ma et al. (2023) integrates known laws with a learnable constitutive model for refinement. SGA Ma et al. (2024) uses LLMs to infer constitutive laws from particle trajectories. However, they rely on labeled data or high-quality motion, which are difficult to acquire. The integration of 3D representation and physical simulation makes it possible to infer intrinsic dynamics from visual observations Xie et al. (2024); Zhong et al. (2024). PAC-NeRF Li et al. (2023) jointly learns NeRF representations and material parameters from multi-view videos. To avoid texture distortion, GIC Cai et al. (2024) presents a geometry supervision framework. PhysDreamer, DreamPhysics, Physics3D, PhysFlow Zhang et al. (2024); Huang et al. (2025); Liu et al. (2024; 2025) guide the estimation process by distilling visual dynamic priors from video diffusion models. However, the parameter estimation process in these methods relies on expert-defined constitutive laws. Spring-Gaus Zhong et al. (2024) integrates a spring-mass system Blickhan (1989) with 3DGS to simulate elastic objects, and optimizes spring stiffness under multi-view video supervision. OmniPhysGS Lin et al. (2025) introduces learnable constitutive Gaussians that assign specific constitutive laws to each Gaussian kernel. enabling interaction simulation in multi-material scenarios. While NeuMA Cao et al. (2024) can learn neural constitutive models Wong et al. (2025) from visual observations, it lacks interpretability and exhibits weak generalization ability. In this paper, we aim to infer constitutive law expressions from visual observations that are both interpretable and highly generalizable.

## D    MATERIAL POINT METHOD

Continuum mechanics studies the deformation and motion behavior of materials under forces. Motion is typically represented by the deformation map $\mathbf{x} = \phi(\mathbf{X}, t)$, which maps from the undeformed material space $\omega^0$ to the deformed world space $\omega^t$. The deformation gradient $\mathbf{F} = \frac{\partial \phi}{\partial \mathbf{X}}(\mathbf{X}, t)$ describes how the material deforms locally. MPM is a simulation method that combines Lagrangian particles with Eulerian grids and has demonstrated its ability to simulate various materials. In MPM, each particle $p$ carries various physical properties, including mass $m$, density $\rho$, volume $V$, Young's modulus $E$, Poisson's ratio $\nu$, velocity $\mathbf{v}$, deformation gradient $\mathbf{F}$ and velocity gradient $\mathbf{C}$. MPM operates within a loop that includes particle-to-grid (P2G) transfer, grid operations, and grid-to-particle (G2P) transfer. In the particle-to-grid (P2G) stage, MPM transfers momentum and mass from particles to grids:

$$m_i^{t+1} = \sum_p w_{ip} m_p, \tag{11}$$

$$(m\mathbf{v})_i^{t+1} = \sum_p w_{ip} \left[ m_p \mathbf{v}_p^t + m_p \mathbf{C}_p^t (\mathbf{x}_i - \mathbf{x}_p^t) \right], \tag{12}$$

where $w_{ip}$ is the B-spline kernel that measures the distance between particle $p$ and grid $i$. After P2G stage, we perform grid operations:

$$\mathbf{v}_i^t = (m\mathbf{v}_i)^t / m_i^t, \tag{13}$$

$$\mathbf{f}_{i,in}^t = -\sum_p \boldsymbol{\tau}_p^t \nabla w_{ip} \mathbf{V}_p, \tag{14}$$

$$\mathbf{v}_i^{t+1} = \mathbf{v}_i^t + \Delta t \left( \mathbf{f}_{i,in} / m_i + \mathbf{g} \right), \tag{15}$$

where $\mathbf{g} = 9.8 \ m/s^2$ denotes the gravitational acceleration. Then we transfer the results back to particles in the grid-to-particle (G2P) stage:

$$\mathbf{v}_p^{t+1} = \sum_i w_{ip} \mathbf{v}_i^{n+1}, \tag{16}$$

$$\mathbf{x}_p^{t+1} = \mathbf{x}_p^t + \Delta t \mathbf{v}_p^{t+1}, \tag{17}$$

$$\mathbf{C}_p^{t+1} = \frac{4}{\Delta \mathbf{x}^2} \sum_i w_{ip} \mathbf{v}_i^{t+1} (\mathbf{x}_i - \mathbf{x}_p^t)^T, \tag{18}$$

$$\mathbf{F}_p^{tr} = \left( \mathbf{I} + \Delta t \mathbf{C}_p^{t+1} \right) \mathbf{F}_p^t, \tag{19}$$

$$\mathbf{F}_p^{t+1} = \varphi_P(\mathbf{F}_p^{tr}), \tag{20}$$

$$\boldsymbol{\tau}_p^{t+1} = \varphi_E(\mathbf{F}_p^{t+1}), \tag{21}$$

where $\varphi_E$ and $\varphi_P$ denote the elastic and plastic constitutive laws, respectively. $F^{tr}$ represents the trial deformation gradient, which is subsequently corrected using the plastic constitutive law $\varphi_P$. $\tau$ denotes the Kirchhoff stress. By following these three stages, we complete a simulation step.

## E    EXPERT-DESIGNED CONSTITUTIVE LAWS

Within the MPM framework, a complete constitutive law consists of an elastic constitutive law and a plastic constitutive law. In our experimental setup, for all scenarios, we initialize the constitutive individual as a combination of a fixed corotated elasticity model and an identity plasticity model. Several well-known classical constitutive laws are presented in the following.

### E.1    ELASTIC CONSTITUTIVE LAW

The elastic constitutive law describes reversible elastic responses of the material under deformation. Here, we use the Kirchhoff stress $\tau$ to express the stress–strain relationship.

### E.1.1 LINEAR ISOTROPIC ELASTICITY.

The Kirchhoff stress is defined as:

$$\boldsymbol{\tau} = \left[ \mu \left( \mathbf{F} + \mathbf{F}^T - 2\mathbf{I} \right) + \lambda \left( \text{tr}(\mathbf{F}) - 3 \right) \mathbf{I} \right] \mathbf{F}^T, \tag{22}$$

where $\mu$ and $\lambda$ are the Lamé parameters.

### E.1.2 FIXED COROTATED ELASTICITY.

The Kirchhoff stress is defined as:

$$\boldsymbol{\tau} = 2\mu \left( \mathbf{F} - \mathbf{R} \right) \mathbf{F}^T + \lambda J \left( J - 1 \right) \mathbf{I}, \tag{23}$$

where $\mathbf{R} = \mathbf{U}\mathbf{V}^T$ and $\mathbf{F} = \mathbf{U}\boldsymbol{\Sigma}\mathbf{V}^T$ is the singular value decomposition of elastic deformation gradient. $J$ is the determinant of $\mathbf{F}$.

### E.1.3 NEO-HOOKEAN ELASTICITY.

The Kirchhoff stress is defined as:

$$\boldsymbol{\tau} = \mu \left( \mathbf{F}\mathbf{F}^T - \mathbf{I} \right) + \lambda \log(J)\mathbf{I}. \tag{24}$$

### E.1.4 STVK ELASTICITY.

The Kirchhoff stress $\boldsymbol{\tau}$ is defined as

$$\boldsymbol{\tau} = \mathbf{U} \left( 2\mu\boldsymbol{\epsilon} + \lambda \, \text{tr}(\boldsymbol{\epsilon})\mathbf{I} \right) \mathbf{V}^T, \tag{25}$$

where $\mathbf{F} = \mathbf{U}\boldsymbol{\Sigma}\mathbf{V}^T$ and $\boldsymbol{\epsilon} = \log(\boldsymbol{\Sigma})$.

### E.2 PLASTIC CONSTITUTIVE LAW

The plastic constitutive law captures irreversible plastic evolution beyond the elastic limit by correcting the trial deformation gradient $\mathbf{F}^{trial}$ to the final deformation gradient $\mathbf{F}$.

### E.2.1 IDENTITY PLASTICITY.

The corrected deformation gradient is defined as:

$$\mathbf{F}^{\text{corrected}} = \mathbf{F} \tag{26}$$

The identity plasticity model does not induce any plastic effects.

### E.2.2 DRUCKER-PRAGER PLASTICITY.

Given $\mathbf{F} = \mathbf{U}\boldsymbol{\Sigma}\mathbf{V}^T$ and $\boldsymbol{\epsilon} = \log(\boldsymbol{\Sigma})$, the corrected deformation gradient is defined as:

$$\mathbf{F}^{\text{corrected}} = \mathbf{U} \, \mathcal{Z}(\boldsymbol{\Sigma}) \, \mathbf{V}^T, \tag{27}$$

$$\mathcal{Z}(\boldsymbol{\Sigma}) = \begin{cases} \mathbf{I}, & \text{if } \text{tr}(\boldsymbol{\epsilon}) > 0, \\ \boldsymbol{\Sigma}, & \text{if } \delta\gamma \leq 0 \text{ and } \text{tr}(\boldsymbol{\epsilon}) \leq 0, \\ \exp\left( \boldsymbol{\epsilon} - \delta\gamma \frac{\hat{\boldsymbol{\epsilon}}}{\|\hat{\boldsymbol{\epsilon}}\|} \right), & \text{otherwise,} \end{cases} \tag{28}$$

Here, $\delta\gamma = \|\hat{\boldsymbol{\epsilon}}\| + \alpha \frac{(d\lambda + 2\mu) \, \text{tr}(\boldsymbol{\epsilon})}{2\mu}$, $\alpha = \sqrt{\frac{2}{3}} \cdot \frac{2 \sin \phi_f}{3 - \sin \phi_f}$ and $\phi_f$ is the friction angle. $\hat{\boldsymbol{\epsilon}} = \text{dev}(\boldsymbol{\epsilon})$. Drucker-Prager plasticity is suitable for simulating materials like snow and sand.

### E.2.3 VON MISES PLASTICITY.

The corrected deformation gradient is defined as:

$$\mathbf{F}^{\text{corrected}} = \mathbf{U}\,\mathcal{Z}(\mathbf{\Sigma})\,\mathbf{V}^T, \tag{29}$$

where

$$\mathcal{Z}(\mathbf{\Sigma}) = \begin{cases} \mathbf{\Sigma}, & \delta\gamma \leq 0, \\ \exp\left(\epsilon - \delta\gamma\frac{\hat{\epsilon}}{\|\hat{\epsilon}\|}\right), & \text{otherwise}, \end{cases} \tag{30}$$

and

$$\delta\gamma = \|\hat{\epsilon}\| - \frac{\tau_Y}{2\mu}. \tag{31}$$

Here $\tau_Y$ is the yield stress. von Mises plasticity is suitable for simulating plasticity like metal and clay.

### E.2.4 FLUID PLASTICITY.

The corrected deformation gradient is defined as:

$$\mathbf{F}^{\text{corrected}} = J^{1/3}\,\mathbf{I}, \tag{32}$$

where $J$ is the determinant of $\mathbf{F}$. Fluid plasticity is suitable for simulating fluid-like materials.

## F LIMITATION AND FUTURE WORK

Although our method effectively captures intrinsic dynamics from visual observations and demonstrates strong interpretability and generalization capabilities, it still has certain limitations that warrant further research and improvement. The method relies on an evolutionary search paradigm that involves extensive evaluations. This process is time-consuming because it requires a large number of forward simulations and backward parameter optimization. Ideally, a preliminary screening mechanism could be introduced, where only individuals with potential merit are subjected to further evaluation. Such a strategy could significantly reduce evaluations and accelerate the efficiency of constitutive law discovery.

## G PROMPT DESIGN DETAILS

In the following, we present the prompts used to guide LLMs to enable the evolution of constitutive laws. To further achieve a decoupled evolution strategy, we designed distinct prompts for the alternating evolution phase and the joint evolution phase.

### G.1 PROMPT DESIGN FOR JOINT EVOLUTION

System prompt:

```
You are an intelligent AI assistant for coding, physical simulation, and scientific discovery.
Follow the user's requirements carefully and make sure you understand them.
Your expertise is strictly limited to physical simulation, material science, mathematics, and
coding.
Keep your answers short and to the point.
Do not provide any information that is not requested.
Always document your code as comments to explain the reason behind them.
Use Markdown to format your solution.
You are very familiar with Python and PyTorch.
Do not use any external libraries other than the libraries used in the examples.
```

User prompt for **elastic and plastic** constitutive law evolution:

```
### Context

This is a physical simulation environment. The physical simulation is built based on the Material
Point Method. The objective of this problem is to fill in a code block so that the result from
executing the code matches the ground-truth result.

The code block defines the full constitutive behavior of the simulated material through two
separate classes:
```

```
1. **PlasticityModel**: defines the deformation gradient correction model. This class contains two
functions that divide the code into a continuous part that defines the differentiable parameters
and a discrete part that defines the symbolic deformation gradient correction model. The input to
the symbolic deformation gradient correction model is the deformation gradient, and the output is
the corrected deformation gradient.
2. **ElasticityModel**: defines the constitutive law that maps corrected deformation gradient to
stress. This class contains two functions that divide the code into a continuous part that defines
the differentiable parameters and a discrete part that defines the symbolic constitutive law. The
input to the symbolic constitutive law is the corrected deformation gradient, and the output is
the Kirchhoff stress tensor.

The simulation applies the `PlasticityModel` first to correct the deformation gradient, then
passes this corrected deformation gradient into the `ElasticityModel` to compute the stress.

States that capture the physical dynamics of the system and metrics that measure the difference
from the ground-truth result are included in the feedback section.

### Task

Look at the following iterations as examples, analyze them, and generate a better solution upon
them.
```

Coding format prompt for **elastic and plastic** constitutive law evolution:

```
### PyTorch Tips
1. When element-wise multiplying two matrix, make sure their number of dimensions match before the
operation. For example, when multiplying `J` (B,) and `I` (B, 3, 3), you should do `J.view(-1, 1,
1)` before the operation. Similarly, `(J - 1)` should also be reshaped to `(J - 1).view(-1, 1,
1)`. If you are not sure, write down every component in the expression one by one and annotate its
dimension in the comment for verification.
2. When computing the trace of a tensor A (B, 3, 3), use `A.diagonal(dim1=1,
dim2=2).sum(dim=1).view(-1, 1, 1)`. Avoid using `torch.trace` or `Tensor.trace` since they only
support 2D matrix.

### Code Requirements

1. The programming language is always python.
2. Annotate the size of the tensor as comment after each tensor operation. For example, `# (B, 3,
3)`.
3. The only library allowed is PyTorch. Follow the examples provided by the user and check the
PyTorch documentation to learn how to use PyTorch.
4. Separate the code into continuous physical parameters that can be tuned with differentiable
optimization and the symbolic constitutive law represented by PyTorch code. Define them
respectively in the `__init__` function and the `forward` function.
5. Always remember the only output of the `forward` function in **PlasticityModel** class is
corrected deformation gradient.
6. Always remember the only output of the `forward` function in **ElasticityModel** class is
Kirchhoff stress tensor, which is defined by the matrix multiplication between the first
Piola-Kirchhoff stress tensor and the transpose of the deformation gradient tensor. Formally, `tau
= P @ F^T`, where tau is the Kirchhoff stress tensor, P is the first Piola-Kirchhoff stress
tensor, and F is the deformation gradient tensor. Do not directly return any other type of stress
tensor other than Kirchhoff stress tensor. Compute Kirchhoff stress tensor using the equation:
`tau = P @ F^T`.
7. The proposed code should strictly follow the structure and function signatures below:

```python
import torch
import torch.nn as nn

class PlasticityModel(nn.Module):

    def __init__(self, param: float = DEFAULT_VALUE):
        """
        Define trainable continuous physical parameters for differentiable optimization.
        Tentatively initialize the parameters with the default values in args.

        Args:
            param (float): the physical meaning of the parameter.
        """
        super().__init__()
        self.param = nn.Parameter(torch.tensor(param))

    def forward(self, F: torch.Tensor) -> torch.Tensor:
        """
        Compute corrected deformation gradient from deformation gradient tensor.

        Args:
            F (torch.Tensor): deformation gradient tensor (B, 3, 3).

        Returns:
            F_corrected (torch.Tensor): corrected deformation gradient tensor (B, 3, 3).
        """
        return F_corrected

class ElasticityModel(nn.Module):

    def __init__(self, param: float = DEFAULT_VALUE):
        """
        Define trainable continuous physical parameters for differentiable optimization.
        Tentatively initialize the parameters with the default values in args.
```

```
        Args:
            param (float): the physical meaning of the parameter.
        """
        super().__init__()
        self.param = nn.Parameter(torch.tensor(param))

    def forward(self, F: torch.Tensor) -> torch.Tensor:
        """
        Compute Kirchhoff stress tensor from deformation gradient tensor.

        Args:
            F (torch.Tensor): deformation gradient tensor (B, 3, 3).

        Returns:
            kirchhoff_stress (torch.Tensor): Kirchhoff stress tensor (B, 3, 3).
        """
        return kirchhoff_stress
```

### Solution Requirements

1. Analyze step-by-step what the potential problem is in the previous iterations based on the
feedback. Think about why the results from previous constitutive laws mismatched with the ground
truth. Do not give advice about how to optimize. Focus on the formulation of the constitutive law.
Start this section with "### Analysis". Analyze all iterations individually, and start the
subsection for each iteration with "#### Iteration N", where N stands for the index. Remember to
analyze every iteration in the history.

2. Think step-by-step what you need to do in this iteration to improve model performance. Consider
both the elasticity and plasticity components.
For the plasticity components:
    Think about if the plasticity is needed to improve performance. Remember that plasticity is
    not necessary. If your analysis supports plasticity, think about how to update deformation
    gradient using plasticity. Think about how to separate your algorithm into a continuous
    physical parameter part and a symbolic deformation gradient correction model part.
For the elasticity components:
    Think about how to separate your algorithm into a continuous physical parameter part and a
    symbolic constitutive law part.
Describe your plan in pseudo-code, written out in great detail. Remember to update the default
values of the trainable physical parameters based on previous optimizations. Start this section
with "### Step-by-Step Plan".

3. Output the code in a single code block "```python ... ```" with detailed comments in the code
block. Do not add any trailing comments before or after the code block. Start this section with
"### Code".

## G.2 PROMPT DESIGN ALTERNATING EVOLUTION

System prompt:

```
You are an intelligent AI assistant for coding, physical simulation, and scientific discovery.
Follow the user's requirements carefully and make sure you understand them.
Your expertise is strictly limited to physical simulation, material science, mathematics, and
coding.
Keep your answers short and to the point.
Do not provide any information that is not requested.
Always document your code as comments to explain the reason behind them.
Use Markdown to format your solution.
You are very familiar with Python and PyTorch.
Do not use any external libraries other than the libraries used in the examples.
```

User prompt for **plastic** constitutive law evolution:

```
### Context

This is a physical simulation environment. The physical simulation is built based on the Material
Point Method. The objective of this problem is to fill in a code block so that the result from
executing the code matches the ground-truth result.

The code block defines the full constitutive behavior of the simulated material through two
separate classes:
1. **PlasticityModel**: defines the deformation gradient correction model. This class contains two
functions that divide the code into a continuous part that defines the differentiable parameters
and a discrete part that defines the symbolic deformation gradient correction model. The input to
the symbolic deformation gradient correction model is the deformation gradient, and the output is
the corrected deformation gradient.
2. **ElasticityModel**: defines the constitutive law that maps corrected deformation gradient to
stress. This class contains two functions that divide the code into a continuous part that defines
the differentiable parameters and a discrete part that defines the symbolic constitutive law. The
input to the symbolic constitutive law is the corrected deformation gradient, and the output is
the Kirchhoff stress tensor.

The simulation applies the `PlasticityModel` first to correct the deformation gradient, then
passes this corrected deformation gradient into the `ElasticityModel` to compute the stress.

States that capture the physical dynamics of the system and metrics that measure the difference
from the ground-truth result are included in the feedback section.
```

```
### Task

In the current task, the ElasticityModel has already been finalized and should remain unchanged.
Please focus exclusively on analyzing and improving the PlasticityModel class. Look at the
following iterations as examples, analyze them, and generate a better plastic constitutive model
based on them.
```

Coding format prompt for **plastic** constitutive law evolution:

```
### PyTorch Tips
1. When element-wise multiplying two matrix, make sure their number of dimensions match before the
operation. For example, when multiplying 'J' (B,) and 'I' (B, 3, 3), you should do 'J.view(-1, 1,
1)' before the operation. Similarly, '(J - 1)' should also be reshaped to '(J - 1).view(-1, 1,
1)'. If you are not sure, write down every component in the expression one by one and annotate its
dimension in the comment for verification.
2. When computing the trace of a tensor A (B, 3, 3), use 'A.diagonal(dim1=1,
dim2=2).sum(dim=1).view(-1, 1, 1)'. Avoid using 'torch.trace' or 'Tensor.trace' since they only
support 2D matrix.

### Code Requirements

1. The programming language is always python.
2. Annotate the size of the tensor as comment after each tensor operation. For example, '# (B, 3,
3)'.
3. The only library allowed is PyTorch. Follow the examples provided by the user and check the
PyTorch documentation to learn how to use PyTorch.
4. Separate the code into continuous physical parameters that can be tuned with differentiable
optimization and the symbolic constitutive law represented by PyTorch code. Define them
respectively in the '__init__' function and the 'forward' function.
5. Always remember the only output of the 'forward' function in **PlasticityModel** class is
corrected deformation gradient.
6. Always remember the only output of the 'forward' function in **ElasticityModel** class is
Kirchhoff stress tensor, which is defined by the matrix multiplication between the first
Piola-Kirchhoff stress tensor and the transpose of the deformation gradient tensor. Formally, 'tau
= P @ F^T', where tau is the Kirchhoff stress tensor, P is the first Piola-Kirchhoff stress
tensor, and F is the deformation gradient tensor. Do not directly return any other type of stress
tensor other than Kirchhoff stress tensor. Compute Kirchhoff stress tensor using the equation:
'tau = P @ F^T'.
7. The proposed code should strictly follow the structure and function signatures below:

'''python
{code}
'''

### Solution Requirements

1. Analyze step-by-step what the potential problem is in the previous iterations based on the
feedback. Think about why the results from previous constitutive laws mismatched with the ground
truth. Do not give advice about how to optimize. Focus on the formulation of the constitutive law.
Start this section with "### Analysis". Analyze all iterations individually, and start the
subsection for each iteration with "#### Iteration N", where N stands for the index. Remember to
analyze every iteration in the history.

2. Think step-by-step what you need to do in this iteration to improve model performance. Consider
both the elasticity and plasticity components.
For the plasticity components:
    Think about if the plasticity is needed to improve performance. Remember that plasticity is
    not necessary. If your analysis supports plasticity, think about how to update deformation
    gradient using plasticity. Think about how to separate your algorithm into a continuous
    physical parameter part and a symbolic deformation gradient correction model part.
For the elasticity components:
    **Do not analyze or modify this part**. Please focus on improving the plastic components.
    Please ensure that the **ElasticityModel** class must remain exactly the same in every
    iteration, and must be reproduced exactly as originally defined.
Describe your plan in pseudo-code, written out in great detail. Remember to update the default
values of the trainable physical parameters based on previous optimizations. Start this section
with "### Step-by-Step Plan".

3. Output the code in a single code block "'''python ... '''" with detailed comments in the code
block. Do not add any trailing comments before or after the code block. Start this section with
"### Code".
```

User prompt for **elastic** constitutive law evolution:

```
### Context

This is a physical simulation environment. The physical simulation is built based on the Material
Point Method. The objective of this problem is to fill in a code block so that the result from
executing the code matches the ground-truth result.

The code block defines the full constitutive behavior of the simulated material through two
separate classes:
1. **PlasticityModel**: defines the deformation gradient correction model. This class contains two
functions that divide the code into a continuous part that defines the differentiable parameters
and a discrete part that defines the symbolic deformation gradient correction model. The input to
the symbolic deformation gradient correction model is the deformation gradient, and the output is
the corrected deformation gradient.
2. **ElasticityModel**: defines the constitutive law that maps corrected deformation gradient to
stress. This class contains two functions that divide the code into a continuous part that defines
```

```
the differentiable parameters and a discrete part that defines the symbolic constitutive law. The
input to the symbolic constitutive law is the corrected deformation gradient, and the output is
the Kirchhoff stress tensor.

The simulation applies the `PlasticityModel` first to correct the deformation gradient, then
passes this corrected deformation gradient into the `ElasticityModel` to compute the stress.

States that capture the physical dynamics of the system and metrics that measure the difference
from the ground-truth result are included in the feedback section.

### Task

In the current task, the PlasticityModel has already been finalized and should remain unchanged.
Please focus exclusively on analyzing and improving the ElasticityModel class. Look at the
following iterations as examples, analyze them, and generate a better elastic constitutive model
based on them.
```

Coding format prompt for **elastic** constitutive law evolution:

```
### PyTorch Tips
1. When element-wise multiplying two matrix, make sure their number of dimensions match before the
operation. For example, when multiplying `J` (B,) and `I` (B, 3, 3), you should do `J.view(-1, 1,
1)` before the operation. Similarly, `(J - 1)` should also be reshaped to `(J - 1).view(-1, 1,
1)`. If you are not sure, write down every component in the expression one by one and annotate its
dimension in the comment for verification.
2. When computing the trace of a tensor A (B, 3, 3), use `A.diagonal(dim1=1,
dim2=2).sum(dim=1).view(-1, 1, 1)`. Avoid using `torch.trace` or `Tensor.trace` since they only
support 2D matrix.

### Code Requirements

1. The programming language is always python.
2. Annotate the size of the tensor as comment after each tensor operation. For example, `# (B, 3,
3)`.
3. The only library allowed is PyTorch. Follow the examples provided by the user and check the
PyTorch documentation to learn how to use PyTorch.
4. Separate the code into continuous physical parameters that can be tuned with differentiable
optimization and the symbolic constitutive law represented by PyTorch code. Define them
respectively in the `__init__` function and the `forward` function.
5. Always remember the only output of the `forward` function in **PlasticityModel** class is
corrected deformation gradient.
6. Always remember the only output of the `forward` function in **ElasticityModel** class is
Kirchhoff stress tensor, which is defined by the matrix multiplication between the first
Piola-Kirchhoff stress tensor and the transpose of the deformation gradient tensor. Formally, `tau
= P @ F^T`, where tau is the Kirchhoff stress tensor, P is the first Piola-Kirchhoff stress
tensor, and F is the deformation gradient tensor. Do not directly return any other type of stress
tensor other than Kirchhoff stress tensor. Compute Kirchhoff stress tensor using the equation:
`tau = P @ F^T`.
7. The proposed code should strictly follow the structure and function signatures below:

```python
{code}
```

### Solution Requirements

1. Analyze step-by-step what the potential problem is in the previous iterations based on the
feedback. Think about why the results from previous constitutive laws mismatched with the ground
truth. Do not give advice about how to optimize. Focus on the formulation of the constitutive law.
Start this section with "### Analysis". Analyze all iterations individually, and start the
subsection for each iteration with "#### Iteration N", where N stands for the index. Remember to
analyze every iteration in the history.

2. Think step-by-step what you need to do in this iteration to improve model performance. Consider
both the elasticity and plasticity components.
For the plasticity components:
    **Do not analyze or modify this part**. Please focus on improving the elastic components.
    Please ensure that the **PlasticityModel** class must remain exactly the same in every
    iteration, and must be reproduced exactly as originally defined.
For the elasticity components:
    Think about how to separate your algorithm into a continuous physical parameter part and a
    symbolic constitutive law part.
Describe your plan in pseudo-code, written out in great detail. Remember to update the default
values of the trainable physical parameters based on previous optimizations. Start this section
with "### Step-by-Step Plan".

3. Output the code in a single code block "```python ... ```" with detailed comments in the code
block. Do not add any trailing comments before or after the code block. Start this section with
"### Code".
```

## H  VISUALIZATION OF INFERRED INTERPRETABLE CONSTITUTIVE LAW

In this section, we show the inferred constitutive laws under different visual scenarios, including
"BouncyBall", "ClayCat", "HoneyBottle", "JellyDuck", "RubberPawn", "SandFish", "Bun" and
"Burger". Since these laws are expressed in the form of Python code snippets, these laws exhibit
strong interpretability and readability, making them easily understandable to humans.

## H.1 BOUNCYBALL

In the BouncyBall scenario, the constitutive law inferred by our method is presented.

```python
import torch
import torch.nn as nn

class PlasticityModel(nn.Module):

    def __init__(self, yield_threshold: float = 0.5):
        """
        Define trainable physical parameter for plasticity yield threshold.
        Initialized to 0.5 to balance plastic effects based on feedback.

        Args:
            yield_threshold (float): logarithmic strain clamp threshold.
        """
        super().__init__()
        self.yield_threshold = nn.Parameter(torch.tensor(yield_threshold))

    def forward(self, F: torch.Tensor) -> torch.Tensor:
        """
        Correct deformation gradient by clamping logarithmic principal strains.

        Args:
            F (torch.Tensor): deformation gradient tensor (B, 3, 3).

        Returns:
            F_corrected (torch.Tensor): corrected deformation gradient tensor (B, 3, 3).
        """
        # SVD of deformation gradient
        U, Sigma, Vh = torch.linalg.svd(F)  # U: (B,3,3), Sigma: (B,3), Vh: (B,3,3)

        # Clamp singular values to avoid numerical problems
        Sigma_clamped = torch.clamp_min(Sigma, 1e-6)  # (B,3)

        # Logarithmic principal strains
        log_sigma = torch.log(Sigma_clamped)  # (B,3)

        # Enforce positive yield threshold via softplus
        yield_thresh = torch.nn.functional.softplus(self.yield_threshold)  # scalar

        epsilon_clamped = torch.clamp(log_sigma, min=-yield_thresh, max=yield_thresh)  # (B,3)

        # Compute corrected singular values
        Sigma_corrected = torch.exp(epsilon_clamped)  # (B,3)

        # Recompose corrected deformation gradient
        F_corrected = torch.matmul(U, torch.matmul(torch.diag_embed(Sigma_corrected), Vh))  # (B
                ,3,3)

        return F_corrected

class ElasticityModel(nn.Module):

    def __init__(self, youngs_modulus_log: float = 10.18, poissons_ratio_sigmoid: float = -0.5):
        """
        Define trainable continuous physical parameters for Corotated Elasticity.

        Args:
            youngs_modulus_log (float): log of Young's modulus.
            poissons_ratio_sigmoid (float): parameter before sigmoid for Poisson's ratio.
        """
        super().__init__()
        self.youngs_modulus_log = nn.Parameter(torch.tensor(youngs_modulus_log))
        self.poissons_ratio_sigmoid = nn.Parameter(torch.tensor(poissons_ratio_sigmoid))

    def forward(self, F: torch.Tensor) -> torch.Tensor:
        """
        Compute Kirchhoff stress tensor from deformation gradient via Corotated Elasticity.

        Args:
            F (torch.Tensor): deformation gradient tensor (B, 3, 3).

        Returns:
            kirchhoff_stress (torch.Tensor): Kirchhoff stress tensor (B, 3, 3).
        """
        B = F.shape[0]

        # Material parameters
        youngs_modulus = self.youngs_modulus_log.exp()  # scalar
        poissons_ratio = self.poissons_ratio_sigmoid.sigmoid() * 0.49  # scalar in (0, 0.49)

        mu = youngs_modulus / (2.0 * (1.0 + poissons_ratio))  # scalar
        la = youngs_modulus * poissons_ratio / ((1.0 + poissons_ratio) * (1.0 - 2.0 *
                poissons_ratio))  # scalar

        # SVD of deformation gradient
        U, Sigma, Vh = torch.linalg.svd(F)  # U: (B,3,3), Sigma: (B,3), Vh: (B,3,3)
```

```
86
87          # Clamp singular values
88          Sigma_clamped = torch.clamp_min(Sigma, 1e-6)  # (B,3)
89
90          # Rotation matrix R
91          R = torch.matmul(U, Vh)  # (B,3,3)
92
93          # Compute determinant
94          J = Sigma_clamped.prod(dim=1).view(B, 1, 1)  # (B,1,1)
95
96          # Identity tensor
97          I = torch.eye(3, device=F.device, dtype=F.dtype).unsqueeze(0).expand(B, 3, 3)  # (B,3,3)
98
99          # Reshape scalars for broadcast
100         mu = mu.view(-1, 1, 1) if mu.dim() == 0 else mu
101         la = la.view(-1, 1, 1) if la.dim() == 0 else la
102
103         # Corotated stress term
104         corotated = 2.0 * mu * (F - R)  # (B,3,3)
105
106         # Volumetric stress term
107         volumetric = la * J * (J - 1).view(B, 1, 1) * I  # (B,3,3)
108
109         # First Piola-Kirchhoff stress tensor P
110         P = corotated + volumetric  # (B,3,3)
111
112         # Kirchhoff stress tau = P @ F^T
113         Ft = F.transpose(1, 2)  # (B,3,3)
114         kirchhoff_stress = torch.matmul(P, Ft)  # (B,3,3)
115
116         return kirchhoff_stress
```

## H.2 CLAYCAT

In the ClayCat scenario, the constitutive law inferred by our method is presented.

```
1  import torch
2  import torch.nn as nn
3
4
5  class PlasticityModel(nn.Module):
6
7      def __init__(self, yield_stress: float = 2.16, shear_modulus: float = 28.0):
8          """
9          Define trainable continuous physical parameters for differentiable optimization.
10         Initialize with best values from iterative feedback.
11
12         Args:
13             yield_stress (float): yield stress threshold for plastic flow.
14             shear_modulus (float): shear modulus for plastic correction.
15         """
16         super().__init__()
17         self.yield_stress = nn.Parameter(torch.tensor(yield_stress))
18         self.shear_modulus = nn.Parameter(torch.tensor(shear_modulus))
19
20     def forward(self, F: torch.Tensor) -> torch.Tensor:
21         """
22         Compute corrected deformation gradient from deformation gradient tensor using von Mises
               plasticity on
23         logarithmic deviatoric principal strains.
24
25         Args:
26             F (torch.Tensor): deformation gradient tensor (B, 3, 3).
27
28         Returns:
29             F_corrected (torch.Tensor): corrected deformation gradient tensor (B, 3, 3).
30         """
31         # SVD of deformation gradient F
32         U, sigma, Vh = torch.linalg.svd(F)  # U: (B,3,3), sigma: (B,3), Vh: (B,3,3)
33         sigma = torch.clamp_min(sigma, 1e-6)  # clamp to prevent log(0), (B,3)
34
35         # Compute principal logarithmic strains
36         epsilon = torch.log(sigma)  # (B,3)
37
38         # Volumetric (mean) strain
39         epsilon_mean = epsilon.mean(dim=1, keepdim=True)  # (B,1)
40
41         # Deviatoric strains
42         epsilon_dev = epsilon - epsilon_mean  # (B,3)
43
44         # Norm of deviatoric strain
45         epsilon_dev_norm = epsilon_dev.norm(dim=1, keepdim=True) + 1e-12  # (B,1)
46
47         # Clamp plasticity parameters to prevent numerical issues
48         yield_stress = torch.clamp_min(self.yield_stress, 1e-6)
49         shear_modulus = torch.clamp_min(self.shear_modulus, 1e-6)
50
51         # Plastic multiplier
52         delta_gamma = epsilon_dev_norm - yield_stress / (2 * shear_modulus)  # (B,1)
```

```
53              delta_gamma_pos = torch.clamp_min(delta_gamma, 0.0)  # (B,1)
54
55              # Correct deviatoric strains by return mapping if yielding
56              epsilon_corrected = epsilon - (delta_gamma_pos / epsilon_dev_norm) * epsilon_dev  # (B,3)
57
58              # Where not yielding, keep original strain
59              yielding_mask = (delta_gamma > 0).view(-1, 1)  # (B,1)
60              epsilon_final = torch.where(yielding_mask, epsilon_corrected, epsilon)  # (B,3)
61
62              # Reconstruct corrected singular values and deformation gradient
63              sigma_corrected = torch.exp(epsilon_final)  # (B,3)
64              diag_sigma_corrected = torch.diag_embed(sigma_corrected)  # (B,3,3)
65
66              F_corrected = torch.matmul(U, torch.matmul(diag_sigma_corrected, Vh))  # (B,3,3)
67
68              return F_corrected
69
70
71  class ElasticityModel(nn.Module):
72
73      def __init__(self, youngs_modulus_log: float = 11.7, poissons_ratio_logit: float = -0.7):
74          """
75          Define trainable continuous physical parameters for differentiable optimization.
76          Initialize with values inferred from analysis.
77
78          Args:
79              youngs_modulus_log (float): log of Young's modulus.
80              poissons_ratio_logit (float): pre-sigmoid parameter for Poisson's ratio.
81          """
82          super().__init__()
83          self.youngs_modulus_log = nn.Parameter(torch.tensor(youngs_modulus_log))
84          self.poissons_ratio_logit = nn.Parameter(torch.tensor(poissons_ratio_logit))
85
86      def forward(self, F: torch.Tensor) -> torch.Tensor:
87          """
88          Compute Kirchhoff stress tensor from deformation gradient tensor using St. Venant-
                  Kirchhoff elasticity.
89
90          Args:
91              F (torch.Tensor): deformation gradient tensor (B, 3, 3).
92
93          Returns:
94              kirchhoff_stress (torch.Tensor): Kirchhoff stress tensor (B, 3, 3).
95          """
96          B = F.shape[0]
97          device = F.device
98          dtype = F.dtype
99
100         # Compute Young's modulus from log
101         youngs_modulus = torch.exp(self.youngs_modulus_log)  # scalar
102
103         # Compute Poisson's ratio from sigmoid(logit) scaled to (0,0.49)
104         poissons_ratio = torch.sigmoid(self.poissons_ratio_logit) * 0.49  # scalar in (0,0.49)
105
106         mu = youngs_modulus / (2 * (1 + poissons_ratio))  # scalar
107         la = youngs_modulus * poissons_ratio / ((1 + poissons_ratio) * (1 - 2 * poissons_ratio))
                  # scalar
108
109         # Identity tensor expanded to batch size
110         I = torch.eye(3, dtype=dtype, device=device).unsqueeze(0).expand(B, -1, -1)  # (B,3,3)
111
112         # Right Cauchy-Green tensor C = F^T F
113         Ft = F.transpose(1, 2)  # (B,3,3)
114         C = torch.matmul(Ft, F)  # (B,3,3)
115
116         # Green-Lagrange strain E = 0.5 * (C - I)
117         E = 0.5 * (C - I)  # (B,3,3)
118
119         # Trace of E computed by summing diagonal elements
120         trE = E.diagonal(dim1=1, dim2=2).sum(dim=1).view(B, 1, 1)  # (B,1,1)
121
122         # Second Piola-Kirchhoff stress tensor S
123         S = 2 * mu * E + la * trE * I  # (B,3,3)
124
125         # First Piola-Kirchhoff stress tensor P = F @ S
126         P = torch.matmul(F, S)  # (B,3,3)
127
128         # Kirchhoff stress tensor tau = P @ F^T
129         kirchhoff_stress = torch.matmul(P, Ft)  # (B,3,3)
130
131         return kirchhoff_stress
```

## H.3  HONEYBOTTLE

In the HoneyBottle scenario, the constitutive law inferred by our method is presented.

```
1  import torch
2  import torch.nn as nn
3
```

```python
class PlasticityModel(nn.Module):

    def __init__(
        self,
        youngs_modulus_log: float = 6.0,
        poissons_ratio_unconstrained: float = -1.0,
        yield_stress: float = 2.5,
    ):
        """
        Plasticity model with logarithmic strain return mapping.

        Args:
            youngs_modulus_log (float): log Young's modulus.
            poissons_ratio_unconstrained (float): unconstrained scalar for Poisson's ratio.
            yield_stress (float): yield stress threshold.
        """
        super().__init__()
        self.youngs_modulus_log = nn.Parameter(torch.tensor(youngs_modulus_log))  # scalar
        self.poissons_ratio_unconstrained = nn.Parameter(torch.tensor(
            poissons_ratio_unconstrained))  # scalar
        self.yield_stress = nn.Parameter(torch.tensor(yield_stress))  # scalar

    def forward(self, F: torch.Tensor) -> torch.Tensor:
        """
        Compute corrected deformation gradient from deformation gradient tensor.

        Args:
            F (torch.Tensor): deformation gradient tensor (B, 3, 3).

        Returns:
            F_corrected (torch.Tensor): corrected deformation gradient tensor (B, 3, 3).
        """
        youngs_modulus = self.youngs_modulus_log.exp()  # scalar
        poissons_ratio = torch.sigmoid(self.poissons_ratio_unconstrained) * 0.49  # scalar in (0,
            0.49)
        yield_stress = self.yield_stress  # scalar

        mu = youngs_modulus / (2.0 * (1.0 + poissons_ratio))

        U, sigma, Vh = torch.linalg.svd(F, full_matrices=False)  # U:(B,3,3), sigma:(B,3), Vh:(B
            ,3,3)

        # Clamp singular values to avoid collapse
        sigma_clamped = torch.clamp_min(sigma, 1e-4)  # (B,3)

        # Logarithmic strain
        epsilon = torch.log(sigma_clamped)  # (B,3)

        # Volumetric strain (trace)
        epsilon_trace = epsilon.sum(dim=1, keepdim=True)  # (B,1)

        # Deviatoric strain
        epsilon_bar = epsilon - epsilon_trace / 3.0  # (B,3)

        # Norm of deviatoric strain (avoid division by zero)
        epsilon_bar_norm = torch.norm(epsilon_bar, dim=1, keepdim=True) + 1e-12  # (B,1)

        # Plastic multiplier
        delta_gamma = epsilon_bar_norm - yield_stress / (2.0 * mu)  # (B,1)

        # Plastic factor (clamped)
        plastic_factor = torch.clamp_min(delta_gamma / epsilon_bar_norm, 0.0)  # (B,1)

        # Correct logarithmic strain
        epsilon_corrected = epsilon - plastic_factor * epsilon_bar  # (B,3)

        # Reconstruct corrected singular values
        sigma_corrected = torch.exp(epsilon_corrected)  # (B,3)

        # Recompose corrected deformation gradient
        F_corrected = torch.matmul(U, torch.matmul(torch.diag_embed(sigma_corrected), Vh))  # (B
            ,3,3)

        return F_corrected

class ElasticityModel(nn.Module):

    def __init__(
        self,
        youngs_modulus_log: float = 11.7,
        poissons_ratio_unconstrained: float = 5.5,
    ):
        """
        Corotated Elasticity model with trainable physical parameters.

        Args:
            youngs_modulus_log (float): log Young's modulus.
            poissons_ratio_unconstrained (float): unconstrained scalar for Poisson's ratio.
        """
        super().__init__()
```

```
 92            self.youngs_modulus_log = nn.Parameter(torch.tensor(youngs_modulus_log))   # scalar
 93            self.poissons_ratio_unconstrained = nn.Parameter(torch.tensor(
                   poissons_ratio_unconstrained))  # scalar
 94
 95        def forward(self, F: torch.Tensor) -> torch.Tensor:
 96            """
 97            Compute Kirchhoff stress tensor from deformation gradient tensor.
 98
 99            Args:
100                F (torch.Tensor): deformation gradient tensor (B, 3, 3).
101
102            Returns:
103                kirchhoff_stress (torch.Tensor): Kirchhoff stress tensor (B, 3, 3).
104            """
105            youngs_modulus = self.youngs_modulus_log.exp()  # scalar
106            poissons_ratio = torch.sigmoid(self.poissons_ratio_unconstrained) * 0.49  # scalar in (0,
                   0.49)
107
108            mu = youngs_modulus / (2.0 * (1.0 + poissons_ratio))
109            la = youngs_modulus * poissons_ratio / ((1.0 + poissons_ratio) * (1.0 - 2.0 *
                   poissons_ratio))
110
111            U, sigma, Vh = torch.linalg.svd(F, full_matrices=False)  # (B,3,3), (B,3), (B,3,3)
112
113            # Clamp singular values for numerical stability
114            sigma_clamped = torch.clamp_min(sigma, 1e-5)  # (B,3)
115
116            # Rotation matrix R = U V^T
117            R = torch.matmul(U, Vh)   # (B,3,3)
118
119            Ft = F.transpose(1, 2)   # (B,3,3)
120
121            # Corotated stress: 2 * mu * (F - R) * F^T
122            corotated_stress = 2.0 * mu * torch.matmul(F - R, Ft)   # (B,3,3)
123
124            # Compute determinant J = product of singular values
125            J = torch.prod(sigma_clamped, dim=1)   # (B,)
126            J = J.view(-1, 1, 1)   # (B,1,1)
127
128            # Identity tensor I
129            I = torch.eye(3, dtype=F.dtype, device=F.device).unsqueeze(0)   # (1,3,3)
130
131            volume_stress = la * J * (J - 1).view(-1, 1, 1) * I   # (B,3,3)
132
133            # First Piola-Kirchhoff stress P
134            P = corotated_stress + volume_stress   # (B,3,3)
135
136            kirchhoff_stress = torch.matmul(P, Ft)   # (B,3,3)
137
138            return kirchhoff_stress
```

## H.4 JELLYDUCK

In the JellyDuck scenario, the constitutive law inferred by our method is presented.

```
 1  import torch
 2  import torch.nn as nn
 3
 4
 5  class PlasticityModel(nn.Module):
 6
 7      def __init__(self, yield_stress: float = 0.1, hardening: float = 0.0):
 8          """
 9          Define trainable continuous physical parameters for differentiable optimization.
10          Initialize yield stress and isotropic hardening parameters.
11
12          Args:
13              yield_stress (float): yield stress threshold for plastic correction.
14              hardening (float): isotropic hardening parameter.
15          """
16          super().__init__()
17          self.yield_stress = nn.Parameter(torch.tensor(yield_stress))   # scalar parameter
18          self.hardening = nn.Parameter(torch.tensor(hardening))         # scalar parameter
19
20      def forward(self, F: torch.Tensor) -> torch.Tensor:
21          """
22          Compute corrected deformation gradient using von Mises plasticity return mapping.
23
24          Args:
25              F (torch.Tensor): deformation gradient tensor (B, 3, 3).
26
27          Returns:
28              F_corrected (torch.Tensor): corrected deformation gradient tensor (B, 3, 3).
29          """
30          B = F.shape[0]
31
32          # SVD of deformation gradient: F = U * diag(sigma) * Vh
33          U, sigma, Vh = torch.linalg.svd(F)   # U,Vh: (B,3,3), sigma: (B,3)
34
```

```python
35          # Clamp singular values to avoid log(0)
36          sigma_clamped = torch.clamp_min(sigma, 1e-5)  # (B, 3)
37
38          # Compute logarithmic strain
39          epsilon = torch.log(sigma_clamped)  # (B, 3)
40
41          # Deviatoric strain: subtract mean (volumetric) strain
42          epsilon_mean = epsilon.mean(dim=1, keepdim=True)  # (B, 1)
43          epsilon_dev = epsilon - epsilon_mean  # (B, 3)
44
45          # Norm of deviatoric strain
46          epsilon_dev_norm = torch.norm(epsilon_dev, dim=1, keepdim=True)  # (B, 1)
47
48          # Effective yield threshold with hardening, clamped to positive
49          yield_threshold = torch.clamp_min(self.yield_stress + self.hardening, 1e-8)  # scalar
50
51          # Plastic correction factor (return mapping)
52          gamma = torch.clamp_min(epsilon_dev_norm - yield_threshold, 0.0) / (epsilon_dev_norm + 1e
                -12)  # (B,1)
53
54          # Correct deviatoric strain
55          epsilon_dev_corrected = epsilon_dev * (1 - gamma)  # (B, 3)
56
57          # Reconstruct corrected logarithmic strain
58          epsilon_corrected = epsilon_dev_corrected + epsilon_mean  # (B, 3)
59
60          # Exponentiate to get corrected singular values
61          sigma_corrected = torch.exp(epsilon_corrected)  # (B, 3)
62
63          # Recompose corrected deformation gradient
64          F_corrected = torch.matmul(U, torch.matmul(torch.diag_embed(sigma_corrected), Vh))  # (B,
                3, 3)
65
66          return F_corrected
67
68
69  class ElasticityModel(nn.Module):
70
71      def __init__(self, youngs_modulus_log: float = 11.49, poissons_ratio_sigmoid: float = 1.00):
72          """
73          Define trainable continuous physical parameters for differentiable optimization.
74          Initialize with previous best values.
75
76          Args:
77              youngs_modulus_log (float): log of Young's modulus.
78              poissons_ratio_sigmoid (float): Poisson's ratio before sigmoid transformation.
79          """
80          super().__init__()
81          self.youngs_modulus_log = nn.Parameter(torch.tensor(youngs_modulus_log))  # scalar
82          self.poissons_ratio_sigmoid = nn.Parameter(torch.tensor(poissons_ratio_sigmoid))  #
                scalar
83
84      def forward(self, F: torch.Tensor) -> torch.Tensor:
85          """
86          Compute Kirchhoff stress tensor using Corotated elasticity model.
87
88          Args:
89              F (torch.Tensor): deformation gradient tensor (B, 3, 3).
90
91          Returns:
92              kirchhoff_stress (torch.Tensor): Kirchhoff stress tensor (B, 3, 3).
93          """
94          B = F.size(0)
95
96          # Recover physical parameters
97          youngs_modulus = self.youngs_modulus_log.exp()  # scalar positive
98          poissons_ratio = self.poissons_ratio_sigmoid.sigmoid() * 0.49  # scalar in [0, 0.49]
99
100         mu = youngs_modulus / (2 * (1 + poissons_ratio))  # (scalar)
101         la = youngs_modulus * poissons_ratio / ((1 + poissons_ratio) * (1 - 2 * poissons_ratio))
                # (scalar)
102
103         # SVD of F
104         U, sigma, Vh = torch.linalg.svd(F)  # (B,3,3), (B,3), (B,3,3)
105         sigma = torch.clamp_min(sigma, 1e-5)  # avoid zero singular values
106
107         # Rotation matrix R = U * Vh
108         R = torch.matmul(U, Vh)  # (B, 3, 3)
109
110         # Determinant J = product of singular values
111         J = torch.prod(sigma, dim=1).view(-1, 1, 1)  # (B, 1, 1)
112
113         # Identity matrix I
114         I = torch.eye(3, dtype=F.dtype, device=F.device).unsqueeze(0).expand(B, -1, -1)  # (B, 3,
                3)
115
116         # Corotated first Piola-Kirchhoff stress: P_corot = 2 * mu * (F - R)
117         mu_expanded = mu.view(-1, 1, 1)  # (B, 1, 1)
118         P_corot = 2 * mu_expanded * (F - R)  # (B, 3, 3)
119
120         # Volume part: P_vol = la * J * (J - 1) * J * F^{-T}
121         F_inv = torch.linalg.inv(F)  # (B, 3, 3)
```

```
122         F_inv_T = F_inv.transpose(1, 2)   # (B, 3, 3)
123         volume_factor = la.view(-1, 1, 1) * J * (J - 1).view(-1, 1, 1)   # (B, 1, 1)
124         P_vol = volume_factor * J * F_inv_T   # (B, 3, 3)
125
126         # Total first Piola-Kirchhoff stress tensor
127         P = P_corot + P_vol   # (B, 3, 3)
128
129         # Kirchhoff stress tensor tau = P @ F^T
130         Ft = F.transpose(1, 2)   # (B, 3, 3)
131         kirchhoff_stress = torch.matmul(P, Ft)   # (B, 3, 3)
132
133         return kirchhoff_stress
```

## H.5 RUBBERPAWN

In the RubberPawn scenario, the constitutive law inferred by our method is presented.

```
1  import torch
2  import torch.nn as nn
3
4
5  class PlasticityModel(nn.Module):
6
7      def __init__(self, yield_stress: float = 0.22, mu_log: float = 4.0):
8          """
9          Define trainable continuous physical parameters for differentiable optimization.
10         Initialize yield_stress and plastic shear modulus (mu) in log space.
11
12         Args:
13             yield_stress (float): yield stress controlling plastic threshold.
14             mu_log (float): log shear modulus for plastic correction.
15         """
16         super().__init__()
17         self.yield_stress = nn.Parameter(torch.tensor(yield_stress))   # scalar
18         self.mu_log = nn.Parameter(torch.tensor(mu_log))   # scalar
19
20     def forward(self, F: torch.Tensor) -> torch.Tensor:
21         """
22         Compute corrected deformation gradient from deformation gradient tensor via logarithmic
                spectral plasticity.
23
24         Args:
25             F (torch.Tensor): deformation gradient tensor (B, 3, 3).
26
27         Returns:
28             F_corrected (torch.Tensor): corrected deformation gradient tensor (B, 3, 3).
29         """
30         B = F.shape[0]
31
32         mu = self.mu_log.exp()   # scalar
33
34         # SVD decomposition
35         U, sigma, Vh = torch.linalg.svd(F)   # U: (B,3,3), sigma: (B,3), Vh: (B,3,3)
36
37         # Clamp singular values
38         sigma = torch.clamp_min(sigma, 1e-6)   # (B,3)
39
40         # Logarithmic principal stretches
41         epsilon = torch.log(sigma)   # (B,3)
42
43         # Compute volumetric mean of epsilon
44         epsilon_mean = epsilon.mean(dim=1, keepdim=True)   # (B,1)
45
46         # Deviatoric log strain
47         epsilon_bar = epsilon - epsilon_mean   # (B,3)
48
49         # Norm of deviatoric strain
50         epsilon_bar_norm = torch.linalg.norm(epsilon_bar, dim=1, keepdim=True)   # (B,1)
51
52         # Plastic multiplier
53         delta_gamma = epsilon_bar_norm - self.yield_stress / (2 * mu)   # (B,1)
54
55         # Clamp to non-negative
56         delta_gamma_clamped = torch.clamp_min(delta_gamma, 0.0)   # (B,1)
57
58         # Avoid division by zero
59         denom = epsilon_bar_norm.clamp_min(1e-8)   # (B,1)
60
61         # Compute correction scale factor
62         scale = 1.0 - delta_gamma_clamped / denom   # (B,1)
63
64         # No correction if yield condition not surpassed
65         scale = torch.where(delta_gamma > 0, scale, torch.ones_like(scale))   # (B,1)
66
67         # Apply correction
68         epsilon_bar_corrected = epsilon_bar * scale   # (B,3)
69
70         # Recompose corrected log strain
71         epsilon_corrected = epsilon_bar_corrected + epsilon_mean   # (B,3)
```

```
72
73          # Inverse log to get corrected singular values
74          sigma_corrected = torch.exp(epsilon_corrected)  # (B,3)
75
76          # Reconstructed corrected deformation gradient
77          F_corrected = U @ torch.diag_embed(sigma_corrected) @ Vh  # (B,3,3)
78
79          return F_corrected
80
81
82  class ElasticityModel(nn.Module):
83
84      def __init__(self, youngs_modulus_log: float = 12.9, poissons_ratio_sigmoid: float = 0.0):
85          """
86          Define trainable continuous physical parameters for differentiable optimization.
87          Initialize parameters from best prior estimates.
88
89          Args:
90              youngs_modulus_log (float): log of Young's modulus.
91              poissons_ratio_sigmoid (float): raw Poisson's ratio parameter before sigmoid scaling.
92          """
93          super().__init__()
94          self.youngs_modulus_log = nn.Parameter(torch.tensor(youngs_modulus_log))  # scalar
95          self.poissons_ratio_sigmoid = nn.Parameter(torch.tensor(poissons_ratio_sigmoid))  #
                scalar
96
97      def forward(self, F: torch.Tensor) -> torch.Tensor:
98          """
99          Compute Kirchhoff stress from corrected deformation gradient tensor using StVK elasticity
                .
100
101         Args:
102             F (torch.Tensor): deformation gradient tensor (B, 3, 3).
103
104         Returns:
105             kirchhoff_stress (torch.Tensor): Kirchhoff stress tensor (B, 3, 3).
106         """
107         B = F.shape[0]
108
109         # Physical parameters
110         youngs_modulus = self.youngs_modulus_log.exp()  # scalar
111
112         # Sigmoid mapping to (0, 0.499) for Poisson's ratio
113         poissons_ratio = torch.sigmoid(self.poissons_ratio_sigmoid) * 0.499  # scalar
114
115         mu = youngs_modulus / (2.0 * (1.0 + poissons_ratio))  # scalar
116         la = youngs_modulus * poissons_ratio / ((1.0 + poissons_ratio) * (1.0 - 2.0 *
                poissons_ratio))  # scalar
117
118         I = torch.eye(3, dtype=F.dtype, device=F.device).unsqueeze(0)  # (1, 3, 3)
119
120         Ft = F.transpose(1, 2)  # (B, 3, 3)
121
122         # Right Cauchy-Green tensor
123         C = torch.matmul(Ft, F)  # (B, 3, 3)
124
125         # Green-Lagrange strain tensor
126         E = 0.5 * (C - I)  # (B, 3, 3)
127
128         # Trace of strain tensor
129         trE = E.diagonal(dim1=1, dim2=2).sum(dim=1).view(B, 1, 1)  # (B, 1, 1)
130
131         # Second Piola-Kirchhoff stress tensor
132         S = 2.0 * mu * E + la * trE * I  # (B, 3, 3)
133
134         # First Piola-Kirchhoff stress tensor
135         P = torch.matmul(F, S)  # (B, 3, 3)
136
137         # Kirchhoff stress tensor: tau = P * F^T
138         kirchhoff_stress = torch.matmul(P, Ft)  # (B, 3, 3)
139
140         return kirchhoff_stress
```

## H.6  SANDFISH

In the SandFish scenario, the constitutive law inferred by our method is presented.

```
1  import torch
2  import torch.nn as nn
3
4
5  class PlasticityModel(nn.Module):
6
7      def __init__(self, yield_stress: float = 0.07):
8          """
9          Define trainable plastic yield stress parameter with enforced numerical stability.
10
11         Args:
12             yield_stress (float): yield stress controlling deviatoric plastic flow magnitude.
```

```
13              """
14              super().__init__()
15              self.yield_stress = nn.Parameter(torch.tensor(yield_stress))
16
17          def forward(self, F: torch.Tensor) -> torch.Tensor:
18              """
19              Compute plasticity-corrected deformation gradient by shrinking deviatoric logarithmic
                    strain.
20
21              Args:
22                  F (torch.Tensor): deformation gradient tensor (B, 3, 3).
23
24              Returns:
25                  F_corrected (torch.Tensor): corrected deformation gradient tensor (B, 3, 3).
26              """
27              # SVD decomposition
28              U, sigma, Vh = torch.linalg.svd(F)                          # (B, 3, 3), (B, 3), (B, 3,
                    3)
29
30              # Clamp singular values for stability
31              sigma_clamped = torch.clamp_min(sigma, 1e-6)                # (B, 3)
32
33              # Compute logarithmic principal strain
34              epsilon = torch.log(sigma_clamped)                          # (B, 3)
35
36              # Volumetric part (mean)
37              epsilon_mean = epsilon.mean(dim=1, keepdim=True)            # (B, 1)
38
39              # Deviatoric strain
40              epsilon_dev = epsilon - epsilon_mean                        # (B, 3)
41
42              # Norm of deviatoric strain
43              epsilon_dev_norm = torch.linalg.norm(epsilon_dev, dim=1, keepdim=True)  # (B, 1)
44
45              # Enforce minimum yield stress to avoid numerical instability
46              yield_stress = torch.clamp_min(self.yield_stress, 0.05)       # scalar
47
48              # Clamp norm for division
49              epsilon_dev_norm_safe = torch.clamp_min(epsilon_dev_norm, 1e-12)        # (B, 1)
50
51              # Compute plastic correction magnitude delta_gamma
52              delta_gamma = epsilon_dev_norm - yield_stress               # (B, 1)
53              delta_gamma_clamped = torch.clamp_min(delta_gamma, 0.0)     # (B, 1)
54
55              # Scaling factor for deviatoric strain correction
56              scale = 1.0 - delta_gamma_clamped / epsilon_dev_norm_safe     # (B, 1)
57              scale = torch.clamp_min(scale, 0.0)                         # (B, 1)
58
59              # Apply plastic correction to deviatoric strain
60              epsilon_dev_corrected = epsilon_dev * scale                 # (B, 3)
61
62              # Recombine volumetric and deviatoric parts
63              epsilon_corrected = epsilon_mean + epsilon_dev_corrected      # (B, 3)
64
65              # Calculate corrected singular values
66              sigma_corrected = torch.exp(epsilon_corrected)              # (B, 3)
67
68              # Reconstruct corrected deformation gradient
69              F_corrected = U @ torch.diag_embed(sigma_corrected) @ Vh      # (B, 3, 3)
70
71              return F_corrected
72
73
74  class ElasticityModel(nn.Module):
75
76      def __init__(self, youngs_modulus_log: float = 9.55, poissons_ratio_sigmoid: float = 2.50):
77          """
78          Define trainable Young's modulus and Poisson's ratio with physically realistic bounds.
79
80          Args:
81              youngs_modulus_log (float): logarithm of Young's modulus.
82              poissons_ratio_sigmoid (float): raw parameter to be passed through sigmoid for
                    Poisson's ratio.
83          """
84          super().__init__()
85          self.youngs_modulus_log = nn.Parameter(torch.tensor(youngs_modulus_log))
86          self.poissons_ratio_sigmoid = nn.Parameter(torch.tensor(poissons_ratio_sigmoid))
87
88      def forward(self, F: torch.Tensor) -> torch.Tensor:
89          """
90          Compute Kirchhoff stress tensor from deformation gradient with corotated elasticity.
91
92          Args:
93              F (torch.Tensor): deformation gradient tensor (B, 3, 3).
94
95          Returns:
96              kirchhoff_stress (torch.Tensor): Kirchhoff stress tensor (B, 3, 3).
97          """
98          B = F.shape[0]
99
100         # Recover material parameters
101         E = self.youngs_modulus_log.exp()                           # scalar
```

```
102        nu_raw = self.poissons_ratio_sigmoid.sigmoid()                # (0,1)
103        nu = nu_raw * 0.45                                            # scale to max 0.45
              Poisson ratio (~stable and compressible)
104
105        mu = E / (2.0 * (1.0 + nu))                                   # scalar
106        lam = E * nu / ((1.0 + nu) * (1.0 - 2.0 * nu))               # scalar
107
108        # Compute SVD
109        U, sigma, Vh = torch.linalg.svd(F)                           # (B, 3, 3), (B, 3), (B, 3,
              3)
110
111        # Clamp singular values to prevent numerical issues
112        sigma_clamped = torch.clamp_min(sigma, 1e-6)                 # (B, 3)
113
114        # Compute rotation part R
115        R = U @ Vh                                                    # (B, 3, 3)
116
117        # Expand mu for broadcasting
118        if mu.dim() > 0:
119            mu_expanded = mu.view(-1, 1, 1)                          # (B, 1, 1)
120        else:
121            mu_expanded = mu                                        # scalar
122
123        # Corotated stress part: 2 * mu * (F - R)
124        corotated_stress = 2.0 * mu_expanded * (F - R)               # (B, 3, 3)
125
126        # Compute determinant J and clamp for stability
127        J = torch.linalg.det(F)                                       # (B,)
128        J_clamped = torch.clamp_min(J, 1e-8)                          # (B,)
129
130        # Identity tensor I (1, 3, 3)
131        I = torch.eye(3, dtype=F.dtype, device=F.device).unsqueeze(0) # (1, 3, 3)
132
133        # Expand and reshape parameters for broadcasting
134        if lam.dim() > 0:
135            lam_expanded = lam.view(-1, 1, 1)                        # (B, 1, 1)
136        else:
137            lam_expanded = lam                                      # scalar
138
139        J_expanded = J_clamped.view(-1, 1, 1)                        # (B, 1, 1)
140        J_minus_1_expanded = (J_clamped - 1.0).view(-1, 1, 1)        # (B, 1, 1)
141
142        # Volumetric stress: lambda * J * (J - 1) * I
143        volumetric_stress = lam_expanded * J_expanded * J_minus_1_expanded * I   # (B, 3, 3)
144
145        # First Piola-Kirchhoff stress
146        P = corotated_stress + volumetric_stress                     # (B, 3, 3)
147
148        # Transpose of deformation gradient
149        Ft = F.transpose(1, 2)                                        # (B, 3, 3)
150
151        # Kirchhoff stress tensor: tau = P @ F^T
152        kirchhoff_stress = P @ Ft                                     # (B, 3, 3)
153
154        return kirchhoff_stress
```

## H.7 Bun

In the Bun scenario, the constitutive law inferred by our method is presented.

```
1  import torch
2  import torch.nn as nn
3
4
5  class PlasticityModel(nn.Module):
6      def __init__(self, yield_stress: float = 0.30):
7          """
8          Trainable continuous yield stress parameter for von Mises plasticity correction.
9
10         Args:
11             yield_stress (float): yield stress threshold for plastic correction.
12         """
13         super().__init__()
14         self.yield_stress = nn.Parameter(torch.tensor(yield_stress))
15
16     def forward(self, F: torch.Tensor) -> torch.Tensor:
17         """
18         Compute corrected deformation gradient from input deformation gradient tensor.
19
20         Args:
21             F (torch.Tensor): deformation gradient tensor (B, 3, 3).
22
23         Returns:
24             F_corrected (torch.Tensor): corrected deformation gradient tensor (B, 3, 3).
25         """
26         # Compute SVD of F: U, sigma, Vh
27         U, sigma, Vh = torch.linalg.svd(F)  # U: (B,3,3), sigma: (B,3), Vh: (B,3,3)
28
29         # Clamp singular values to avoid log(0)
```

```python
30          sigma_clamped = torch.clamp_min(sigma, 1e-6)  # (B,3)
31
32          # Compute logarithm of singular values (principal logarithmic strains)
33          epsilon = torch.log(sigma_clamped)  # (B,3)
34
35          # Compute volumetric mean strain
36          epsilon_mean = epsilon.mean(dim=1, keepdim=True)  # (B,1)
37
38          # Deviatoric strain (deviation from mean)
39          epsilon_dev = epsilon - epsilon_mean  # (B,3)
40
41          # Norm of deviatoric strain, clamp to avoid numerical issues
42          epsilon_dev_norm = torch.norm(epsilon_dev, dim=1, keepdim=True).clamp_min(1e-12)  # (B,1)
43
44          # Compute plastic multiplier (excess over yield stress)
45          delta_gamma = epsilon_dev_norm - self.yield_stress  # (B,1)
46
47          # Apply plastic correction only if exceeding yield stress
48          delta_gamma_clamped = torch.clamp_min(delta_gamma, 0.0)  # (B,1)
49
50          # Calculate shrink factor for deviatoric strains
51          shrink_factor = 1.0 - delta_gamma_clamped / epsilon_dev_norm  # (B,1)
52
53          # Correct deviatoric strain by projecting onto yield surface
54          epsilon_dev_corrected = epsilon_dev * shrink_factor  # (B,3)
55
56          # Reassemble corrected total logarithmic strains
57          epsilon_corrected = epsilon_mean + epsilon_dev_corrected  # (B,3)
58
59          # Exponentiate to get corrected singular values
60          sigma_corrected = torch.exp(epsilon_corrected)  # (B,3)
61
62          # Reconstruct corrected deformation gradient: F_corrected = U * diag(sigma_corrected) *
                Vh
63          F_corrected = U @ torch.diag_embed(sigma_corrected) @ Vh  # (B,3,3)
64
65          return F_corrected
66
67
68  class ElasticityModel(nn.Module):
69      def __init__(self, youngs_modulus_log: float = 9.82, poissons_ratio_sigmoid: float = 4.07):
70          """
71          Trainable continuous parameters for Neo-Hookean elasticity.
72
73          Args:
74              youngs_modulus_log (float): log of Young's modulus.
75              poissons_ratio_sigmoid (float): Poisson's ratio parameter before sigmoid scaling.
76          """
77          super().__init__()
78          self.youngs_modulus_log = nn.Parameter(torch.tensor(youngs_modulus_log))
79          self.poissons_ratio_sigmoid = nn.Parameter(torch.tensor(poissons_ratio_sigmoid))
80
81      def forward(self, F: torch.Tensor) -> torch.Tensor:
82          """
83          Compute Kirchhoff stress tensor from deformation gradient tensor using Neo-Hookean
                elasticity.
84
85          Args:
86              F (torch.Tensor): deformation gradient tensor (B, 3, 3).
87
88          Returns:
89              kirchhoff_stress (torch.Tensor): Kirchhoff stress tensor (B, 3, 3).
90          """
91          B = F.size(0)  # batch size
92
93          # Compute Young's modulus E and Poisson's ratio nu
94          E = self.youngs_modulus_log.exp()  # scalar
95          nu = self.poissons_ratio_sigmoid.sigmoid() * 0.49  # scalar in (0,0.49)
96
97          mu = E / (2 * (1 + nu))  # scalar
98          lam = E * nu / ((1 + nu) * (1 - 2 * nu))  # scalar
99
100         # Identity tensor I (B,3,3)
101         I = torch.eye(3, dtype=F.dtype, device=F.device).unsqueeze(0).expand(B, -1, -1)  # (B
                ,3,3)
102
103         # Compute determinant J of F (B,)
104         J = torch.linalg.det(F).clamp_min(1e-12).view(-1, 1, 1)  # (B,1,1)
105         logJ = torch.log(J)  # (B,1,1)
106
107         # Compute inverse transpose of F (B,3,3)
108         F_inv = torch.inverse(F)  # (B,3,3)
109         F_inv_T = F_inv.transpose(1, 2)  # (B,3,3)
110
111         # Compute first Piola-Kirchhoff stress tensor P = mu*(F - F_inv_T) + lam*logJ*F_inv_T
112         P = mu * (F - F_inv_T) + lam * logJ * F_inv_T  # (B,3,3)
113
114         # Compute Kirchhoff stress tau = P * F^T
115         Ft = F.transpose(1, 2)  # (B,3,3)
116         kirchhoff_stress = torch.matmul(P, Ft)  # (B,3,3)
117
118         return kirchhoff_stress
```

## H.8    BURGER

In the Burger scenario, the constitutive law inferred by our method is presented.

```python
import torch
import torch.nn as nn

class PlasticityModel(nn.Module):

    def __init__(self):
        """
        Identity plasticity: no correction to deformation gradient.
        """
        super().__init__()

    def forward(self, F: torch.Tensor) -> torch.Tensor:
        """
        Args:
            F (torch.Tensor): deformation gradient tensor (B, 3, 3).

        Returns:
            F_corrected (torch.Tensor): corrected deformation gradient tensor (B, 3, 3).
        """
        # No plastic correction
        return F  # (B, 3, 3)

class ElasticityModel(nn.Module):

    def __init__(self,
                 youngs_modulus_log: float = 8.37,
                 poissons_ratio: float = 0.49):
        """
        Corotated elasticity with trainable parameters.

        Args:
            youngs_modulus_log (float): log of Young's modulus.
            poissons_ratio (float): Poisson's ratio (clamped [0,0.49]).
        """
        super().__init__()
        self.youngs_modulus_log = nn.Parameter(torch.tensor(youngs_modulus_log))
        self.poissons_ratio = nn.Parameter(torch.tensor(poissons_ratio))

    def forward(self, F: torch.Tensor) -> torch.Tensor:
        """
        Compute Kirchhoff stress tensor from deformation gradient tensor.

        Args:
            F (torch.Tensor): deformation gradient tensor (B, 3, 3).

        Returns:
            kirchhoff_stress (torch.Tensor): Kirchhoff stress tensor (B, 3, 3).
        """
        B = F.shape[0]

        # Physical parameters
        E = self.youngs_modulus_log.exp()  # scalar
        nu = torch.clamp(self.poissons_ratio, 0.0, 0.49)  # scalar

        mu = E / (2.0 * (1.0 + nu))  # scalar
        la = E * nu / ((1.0 + nu) * (1.0 - 2.0 * nu))  # scalar

        # SVD of F: U, Sigma, Vh such that F = U @ diag(Sigma) @ Vh
        U, sigma, Vh = torch.linalg.svd(F)  # U: (B,3,3), sigma: (B,3), Vh: (B,3,3)
        sigma = torch.clamp_min(sigma, 1e-5)  # (B,3) ensure positivity

        # Rotation R = U @ Vh
        R = torch.matmul(U, Vh)  # (B,3,3)

        # Corotated stress part: tau_c = 2*mu*(F - R) @ F^T
        Ft = F.transpose(1, 2)  # (B,3,3)
        tau_c = 2.0 * mu * torch.matmul(F - R, Ft)  # (B,3,3)

        # Volumetric part: tau_v = lambda * J * (J - 1) * I
        J = torch.prod(sigma, dim=1).view(B, 1, 1)  # (B,1,1)
        I = torch.eye(3, dtype=F.dtype, device=F.device).unsqueeze(0).expand(B, -1, -1)  # (B
            ,3,3)
        tau_v = la * J * (J - 1) * I  # (B,3,3)

        # Kirchhoff stress
        kirchhoff_stress = tau_c + tau_v  # (B,3,3)

        return kirchhoff_stress
```

