# OpenReview forum: "VisionLaw: Inferring Interpretable Intrinsic Dynamics from Visual Observations via Bilevel Optimization"
_ICLR.cc/2026/Conference — ICLR 2026 Poster_

### Official Review · Reviewer_cSP2 · 2025-10-30

**Soundness:** 3
**Presentation:** 3
**Contribution:** 3
**Rating:** 6
**Confidence:** 3

**Summary:**

This paper proposes VisionLaw, a bilevel framework that uses LLM-driven search to generate/edit symbolic elastic and plastic constitutive laws (upper level) and a differentiable MPM+renderer loop to fit continuous material parameters from video supervision (lower level). The experiments report the best average Chamfer distance on synthetic data, and improved visual fidelity/generalization vs baselines.

**Strengths:**

1. The idea of combining LLM-based hypothesis search with differentiable, vision-guided evaluation to discover interpretable constitutive laws directly from videos is very compelling.
2. Implementation details are transparent and the source code is provided, largely promoting reproducibility.
3. The paper is clearly written and easy to understand.
4. The comprehensive experimental results are solid, proving the effectiveness very well.

**Weaknesses:**

1. While the decoupled strategy is motivated, the paper could quantify search efficiency (e.g., wall-clock, simulator calls, accepted offspring) and compare against naive joint search under equal budgets.
2. The pipeline depends on accurate camera parameters and a 3DGS reconstruction from the first frame. Its sensitivity to calibration errors, occlusions, and reconstruction artifacts is not analyzed.

**Questions:**

Do the learned laws transfer across different objects with similar physical properties in real scenes without re-search?

---

> ### Author Response · Authors · 2025-11-21
>
> We highly appreciate your acknowledgment and encouragement toward our work. Your comments are highly valuable to us.
>
> For W1:
>
> **Table 1:** **Ablation Study on Decoupled Evolution Strategy**. Comparison between naive joint optimization (w/o decouple) and the proposed decoupled strategy (w/ decouple). Left values indicate RGB loss (lower is better); right values indicate average time consumption (in seconds).
>
> | Methods      | BouncyBall        | ClayCat            | HoneyBottle        | Jellyduck      | RuberPawn       | SandFish       | Average         |
> | ------------ | ----------------- | ------------------ | ------------------ | -------------- | --------------- | -------------- | --------------- |
> | w/o decouple | 9.08/11790        | 17.31/10997        | 3.64/10891         | 6.24/**9333**  | 25.88/**10605** | 2.24/**14439** | 10.73/**11343** |
> | w/ decouple  | **7.80**/**9319** | **8.13**/**10636** | **3.54**/**10699** | **5.93**/11249 | **15.25**/10839 | **2.17**/15689 | **7.14**/11405  |
>
> Ablation Study Setup for Decoupled Strategy:
>
> - **Decoupled strategy**: Performs **4 alternating evolution iterations** (2 plastic evolution iterations + 2  elastic evolution iterations), followed by **1 joint optimization iteration**,  totaling **5 upper-level iterations**.
> - **Naive joint search (without decoupling)**: Performs **5  joint optimization iterations**, also totaling **5 upper-level iterations**.
> - **Lower-level optimization**: As described in the paper, both strategies run **10 iterations of gradient optimization** to fine-tune the material properties for each constitutive law.
> - To mitigate randomness, **five independent runs** were conducted using different random seeds.
>
> **Simulator calls**:
> In each upper-level iteration, **18 offspring** are generated, and each constitutive individual undergoes **10 iterations** of lower-level material parameter optimization. Therefore, both strategies result in a total of: **5 (upper-level iterations) \* 18 (offspring) \* 10 (lower-level iterations) = 900 simulator calls**.
>
> **Accepted offspring**:
> During alternating optimization, **3 top offspring** are selected to generate the next **18  offspring**; during joint optimization, **5 top offspring** are selected to generate the next **18 offspring**.
>
> **Results Presentation**:
>
> - **Table 1** reports a comparison between the decoupled strategy and the naive joint search, evaluating both **accuracy** (in terms of optimal RGB loss) and **average time consumption** across different random seed setups.
> - **4.3.3 ABLATION STUDY ON DECOUPLED EVOLUTION STRATEGY** in the paper compares the solutions in terms of **accuracy** and **diversity** for both strategies under the same experimental setting.
>
> These results collectively demonstrate that the decoupled strategy, **with similar computational resource consumption**, significantly **improves the accuracy and diversity** of solutions, more effectively leveraging the LLM's capability for constitutive discovery.
>
> For W2:
>
> Thank you for your valuable consideration. **Indeed**, our work depends on accurate camera parameters and 3DGS reconstruction from the first frame. Calibration errors, occlusions, and reconstruction artifacts could cause persistent discrepancies between the simulated visual dynamics and real visual observations. These discrepancies may, in turn, lead to inherent biases between the inferred intrinsic dynamics and the true underlying dynamics. We place great importance on this issue, and plan to explore and address this challenge in future work. Thank you for your insightful suggestion!
>
> For Q1:
>
> Yes, our method enables the transfer of  the learned laws to different objects with similar physical properties in real scenes **without re-search**. This capability is effectively demonstrated in **Section 4.3.2 GENERALIZATION TO NOVEL SCENARIOS** and further supported by the newly added **Appendix B.5 GENERALIZATION COMPARISON** in the paper. We directly applied the intrinsic dynamics learned from synthetic datasets to a new scene, generating **4D interactive dynamics** that closely match the original observations. This holds great potential to advance physics-driven 4D interaction, embodied intelligence, and virtual reality!

---

> > ### Comment · Reviewer_cSP2 · 2025-11-25
> >
> > Thanks for your response. I have no further concerns.

---

> > > ### Author Response · Authors · 2025-11-26
> > > **Acknowledgments**
> > >
> > > We sincerely appreciate the reviewer’s valuable comments and encouraging feedback. We are very grateful for the recognition of our work. Thank you!

---

### Official Review · Reviewer_9Waz · 2025-11-02

**Soundness:** 2
**Presentation:** 2
**Contribution:** 2
**Rating:** 2
**Confidence:** 5

**Summary:**

The paper propose a framework for evolving symbolic constitutive models and optimizing material parameters from multi-view videos. The task setting follows PAC-NeRF and use PhysGaussian-based simulator for better reconstruction quality. The main contribution compared to PAC-NeRF is its LLM-based constitutive law evolution framework: the non-differentiable discrete constitutive class optimization is done by LLM code generation based on feedbacks from the lower-level continuous physical parameter optimization. The lower-level optimization uses differentiable simulation and differentiable rendering, similar to PAC-NeRF.

**Strengths:**

- LLM-based code generation is a promising way to handle discrete constitutive models, as it can reduce the need for mechanics experts to manually set up these models.

**Weaknesses:**

- Since the upper-level optimization is the main contribution, a more thorough examination is needed:
    - How many upper-level evolutions are need to converge to the correct one?
    - How are different elasticity models distinguished by LLM? Since many models behave similarly at small deformation magnitudes. I don't think LLM can differentiate StVK, neo-Hookean, fixed-corotated when the object is pure elastic.
    - What if the LLM outputs python code with bugs that can crush the program?
    - Why does direct joint evolution not perform well? It appears that the difference lies primarily in prompting and the available information and feedbacks are the same. Ablation studies are needed to motivate decoupled evolution.


- Although the high-level ideas are clear, some details are lacking for reproducibility:
    - What (modalities) are included in the feedback from the lower-level optimization?
    - How is the first batch of constitutive model candidates are initialized? Are they generated by LLM based solely on multiview videos?
    - How are the initial physical parameters set when the predicted constitutive models change?

**Questions:**

- The plastic corrections are derived from plastic flows applied on a specific elastic model. That is, even with the same plastic flow, different elastic model can result in different discrete deformation corrections (called plastic return mappings). For example, the von-Mises plasticity and Drucker-Prager plasticity used in the paper assume stVK elasticity model. The combination of fixed-corotated elasticity and von-Mises plasticity in H.1 BOUNCYBALL is fundamentally wrong.

- How many lower optimizations are expected to run? It seems that each upper-level optimization are followed with multiple independent lower optimizations. Considering the above combination constraints, there are not many available constraints, for example, fewer than 10. I think it is doable to run brute-force lower-level optimizations to find the most fit, i.e., PAC-NeRF + brute-force sweep. This could be a strong baseline to compare running time.

---

> ### Author Response · Authors · 2025-11-21
> **Response to Weakness**
>
> Thank you for your insightful comments, which are greatly valued by us. We will address your concerns below.
>
> For W1:
>
> Thank you for your question. As stated in **Appendix A.1 IMPLEMENTATION DETAILS**, in all experimental settings, we fix the number of upper-level evolution to **7 iterations**, including **4 alternating-evolution steps** and **3 joint-evolution steps**.
>
> For W2:
>
> Thank you for your question. We first clarify that **our goal is to use LLMs to search for constitutive laws that can reconstruct the observed visual dynamics through simulation**—not to have LLMs directly “recognize” or “classify” specific models like StVK, neo-Hookean, or fixed-corotated from visual observations. Importantly, the LLMs leverage its knowledge of existing constitutive models (e.g., StVK, neo-Hookean, fixed-corotated) to **heuristically construct, reformulate, and discover new constitutive forms** that better explain the underlying dynamics in the visual observations.
>
> For W3:
>
> Thank you for your insightful observation. We recognize that LLMs may occasionally generate Python code with bugs, typically due to ill-posed constitutive hypotheses that result in numerical crashes. To address this, we implement a **subprocess isolation mechanism**: each hypothesis is evaluated in an independent subprocess. If a crash occurs, the main process catches the exception and assigns the hypothesis with a poor fitness score (e.g., infinity), treating it as invalid and excluding it from further evolution. Implementation details are provided in the submitted source code included in the supplementary materials.
>
> For W4:
>
> **Table 1:** **Ablation Study on Decoupled Evolution Strategy**. Comparison between naive joint optimization (w/o decouple) and the proposed decoupled strategy (w/ decouple). Left values indicate RGB loss (lower is better); right values indicate time consumption (in seconds).
>
> | Methods      | BouncyBall        | ClayCat            | HoneyBottle        | Jellyduck      | RuberPawn       | SandFish       | Average         |
> | ------------ | ----------------- | ------------------ | ------------------ | -------------- | --------------- | -------------- | --------------- |
> | w/o decouple | 9.08/11790        | 17.31/10997        | 3.64/10891         | 6.24/**9333**  | 25.88/**10605** | 2.24/**14439** | 10.73/**11343** |
> | w/ decouple  | **7.80**/**9319** | **8.13**/**10636** | **3.54**/**10699** | **5.93**/11249 | **15.25**/10839 | **2.17**/15689 | **7.14**/11405  |
>
> Thank you for your detailed observation. A complete constitutive model consists of both elastic and plastic components. Optimizing both parts jointly **greatly increases the search space**, making it more **difficult for the LLM to converge to a high-quality solution**. To address this, we adopt a decoupled evolution strategy, which decomposes the search into simpler sub-tasks, thereby narrowing the search space and improving optimization efficiency. As shown in **Section 4.3.3 ABLATION STUDY ON DECOUPLED EVOLUTION STRATEGY** and the newly added **Table 1**, the results show that the decoupled strategy significantly improves both accuracy and diversity without additional computational overhead.
>
> For W5:
>
> Thank you for your question. As stated in **Section 3.2 LOWER-LEVEL CONSTITUTIVE EVALUATION**, lines 287-288, the feedback primarily includes the following:
>
> - **Material parameter update trajectory**: Tracks the material parameter changes at each iteration during the lower-level optimization.
> - **RGB loss curve**: Tracks the RGB loss at each iteration during the lower-level optimization.
>
> Here is an example of the specific feedback information:
>
> ```
> ### Feedback
> #### Physical parameter training curves (versus iteration)
>
> - elasticity/poissons_ratio_sigmoid: [-0.30, -0.18, -0.18, -0.24, -0.28, -0.28, -0.29, -0.31, -0.33, -0.34, -0.34] (Best: -0.33)
> - elasticity/youngs_modulus_log: [10.09, 10.15, 10.22, 10.18, 10.15, 10.18, 10.20, 10.18, 10.17, 10.18, 10.19] (Best: 10.17)
> - plasticity/yield_stress_log: [0.22, 0.31, 0.37, 0.40, 0.39, 0.35, 0.31, 0.28, 0.27, 0.28, 0.28] (Best: 0.27)
>
> #### Loss training curves (versus iteration)
>
> - rgb (Key loss): [9.3261, 10.8463, 8.9456, 9.5561, 8.8215, 9.1021, 8.6693, 8.7385, 8.6316, 8.7302, 8.6597] (Best: 8.6316)
> ```
>
> For W6:
>
> Thank you for your question. As stated in **Section 4.1.1 IMPLEMENTATION DETAILS** and **Appendix A.1 IMPLEMENTATION DETAILS**, the **initial constitutive model candidate in all scenarios** is only defined as **a purely elastic model**, consisting of fixed corotated elasticity with identity plasticity (the specific forms can be found in Appendix E.1.1 FIXED COROTATED ELASTICITY and E.2.1 IDENTITY PLASTICITY).
>
> For W7:
>
> Thank you for your question. When the constitutive model changes, the LLMs will refer to the material parameter update trajectory from the feedback (see W5 response) to **directly set the initial physical parameters**.

---

> ### Author Response · Authors · 2025-11-21
> **Response to Questions**
>
> For Q1:
>
> Thank you for your question. I'll do my best to address your concerns. In this paper, we do not select and combine existing constitutive models (such as von-Mises plasticity, Drucker-Prager plasticity, stVK elasticity, etc.), as these conventional models fails to capture the full diversity of real-world dynamic behaviors. Instead, we leverage the LLM's extensive constitutive prior to **construct, reformulate, and discover new constitutive forms** that more effectively represent the underlying dynamics of visual observations.
>
> Furthermore, it is worth emphasizing that the plasticity model presented in **Appendix H.1 BOUNCYBALL** is not conventional von-Mises plasticity. Rather, it is a **new constitutive form** constructed by the LLMs during the upper-level evolution. I hope this clarifies your concerns. If you have any further questions, feel free to continue the discussion. Thank you!
>
> For Q2:
>
> Thank you for your question. I'll do my best to address your concerns. As stated in **Appendix A.1 IMPLEMENTATION DETAILS**, the lower-level optimization is set to run for **10 iterations** across all experimental scenarios.
>
> It is important to clarify that the goal of the upper-level optimization is **not** to combine existing constitutive models, as such models fall short in capturing the full diversity of real-world dynamic behaviors. Instead, we leverage the LLM's extensive constitutive prior to **construct, reformulate, and discover new constitutive forms** that more effectively capture the underlying dynamics of visual observations. Given this context, the search space for constitutive models is infinite, making brute-force searches infeasible.

---

### Official Review · Reviewer_aJkm · 2025-11-03

**Soundness:** 3
**Presentation:** 3
**Contribution:** 2
**Rating:** 4
**Confidence:** 4

**Summary:**

This work, titled VisionLaw, proposes a bilevel framework for the visual grounding of physical laws. At the upper level, the Vision-Language Model (VLM) generates a hypothesis about the physical materials. At the lower level, a differentiable simulation-and-rendering pipeline verifies this hypothesis. The main contribution claimed is the bilevel framework for symbolic estimation. The authors evaluate VisionLaw on real and synthetic datasets.

**Strengths:**

- The presentation is straightforward and easy to understand.

- The discussed topic—inferring interpretable intrinsic physics—is fascinating.

- The experiments conducted are thorough and comprehensive.

**Weaknesses:**

Please take the time to read the following two papers:
[1] "LLM and Simulation as Bilevel Optimizers: A New Paradigm to Advance Physical Scientific Discovery."
[2] "Neuma: Neural Material Adaptor for Visual Grounding of Intrinsic Dynamics."

In comparison to these papers, one concern I have is that the contribution of the current work seems weak. The bilevel framework closely resembles that of Paper [1] (which also uses a bilevel framework to search optimal material parameters), and the vision-guided constitutive evaluation mechanism appears to be based on concepts from Paper [2].

**Questions:**

I believe the author should clarify the difference between [1,2] and their work. If this concern is addressed, I will consider changing my score.

---

> ### Author Response · Authors · 2025-11-21
>
> Thank you for your insightful comments, which are greatly valued by us. We will address your concerns below.
>
> **Table 1** clearly highlights the key differences between *VisionLaw* and existing works.
>
>
>
> **Differences from SGA [1]:**
>
> * **Lack of Visual Physical Reasoning**
>
> SGA requires particle trajectories of objects as observations. However, accurate particle-level motion trajectories are inaccessible in real scenes, which makes SGA difficult to apply to real-world problems.
>
> ***To overcome this limitation***:
>
> We first unify constitutive evolution and vision-guided constitutive evaluation, achieving the intrinsic dynamics inference from readily accessible visual observations alone. **Section 4.2.2 REAL-WORLD DATASET** clearly demonstrates our ability to infer interpretable underlying dynamics from real visual observations.
>
> * **Inability to infer complete constitutive laws**
>
> A complete constitutive model consists of both elastic and plastic components; jointly searching over both leads to an enormous search space and poor convergence. As a result, SGA can only infer elasticity given known plasticity, or infer plasticity given known elasticity—thus fundamentally lacking the ability to discover a complete constitutive law.
>
> ***To overcome this limitation***:
>
> We propose a decoupled evolution strategy, which can effectively alleviate the search space explosion caused by jointly evolving elastic and plastic components, thereby markedly improving both search efficiency and solution quality of discovering complete constitutive models.
>
>
>
> **Difference from NeuMA [2]:**
>
> * **Lack of interpretability**
>
> NeuMA jointly models elastic and plastic behavior from visual observations using neural networks. However, their black-box nature limits their interpretability for both LLMs and humans, hindering integration with LLMs and constraining their value for scientific discovery.
>
> * **Lack of physical inductive bias**
>
> Due to the lack of physical inductive biases and the much sparser nature of visual supervision compared with particle-level supervision, NeuMA tends to “mechanically fit” the visual observations rather than capture the underlying dynamics, resulting in overfitting and poor generalization.
>
> ***To overcome the two limitations mentioned above:***
>
> We propose *VisionLaw* that unifies LLMs-driven constitutive evolution with vision-guided constitutive evaluation, enabling the discovery of symbolic constitutive laws from visual observations. The integration of LLMs implicitly introduces physical inductive biases, allowing our approach to better capture the underlying dynamics, rather than merely reconstructing appearances, thereby achieving strong generalization.
>
> To further validate the superior generalization of our method over NeuMA, we **provide additional comparative experiments** on a 4D interaction generation task. Detailed results are presented in **Appendix B.5 GENERALIZATION COMPARISON**. We observe that NeuMA frequently struggles when transferred to new scenes, exhibiting *numerical instabilities* and *incorrect dynamics*. In contrast, VisionLaw consistently generates stable 4D interactive dynamics that match the original observations.
>
>
>
> **Differences from SGA [1] and NeuMA [2]:**
>
> * **Inapplicability to non-differentiable simulation scenarios**
>
>   This aspect remains unexplored in SGA. NeuMA requires optimizing a neural constitutive model to match observations. Because the network contains a large number of parameters, it is nearly impossible to optimize without gradients, making NeuMA unsuitable for non-differentiable simulation scenarios.
>
> To demonstrate the potential of VisionLaw in non-differentiable simulation environments, we **provide additional experiments to compare two types of lower-level parameter optimization**: (i) the gradient-based optimization used as the default in this work, and (ii) gradient-free evolutionary strategies implemented via the differential evolution algorithm. Please refer to **Section 4.3.4 POTENTIAL ANALYSIS IN NON-DIFFERENTIABLE SIMULATION ENVIRONMENTS** and **the above Table 2** for detailed experiments. The results show that VisionLaw remains effective in inferring intrinsic dynamics even when the lower-level optimization uses gradient-free evolutionary strategies, highlighting its strong potential to extend from differentiable simulation environments to non-differentiable ones.

---

### Official Review · Reviewer_HDyU · 2025-11-04

**Soundness:** 4
**Presentation:** 2
**Contribution:** 4
**Rating:** 8
**Confidence:** 3

**Summary:**

The paper proposes a bilevel optimization framework for estimating the physics of a scene. This is done with an LLm at the high level iteratively refining the physical formulas governing the particles in the scene (through an evolutionary algorith) and a low-level differentiable particle-based simulator refining the individual particle properties.

**Strengths:**

- **S.1:** Great idea. I think this is a really beautiful and elegant idea. Having the physics defined in the outer loop and the parameters tuned in the inner loop is great.
- **S.2:** Clear Results. As far as I can tell, the results look pretty impressive and the paper compares its method to a variety of contemporary baselines, which is great.
- **S.3:** Reproducibility. I appreciate that the authors released their source code, additional videos, and all LLM prompts.

**Weaknesses:**

- **W.1:** The writing is incredibly dense. I've worked in parameter identification for simulator tuning and I've written my own physics engines and I was barely able to follow all of this. A bit simpler writing would greatly benefit this paper. For example, it took me a while to understand why you're not just using the gradient that you get from the differentiable simulator rollout to also tune the high-level physics.

**Questions:**

- **Q.1:** More of a comment: Your contributions are weirdly written. I think contibution 1 is actually made up of of contibutions 2 and 3. So that should be one bullet point. And running experiments that validate your method is not a contribution, so contribution 4 should be removed.
- **Q.2:** Another suggestion: since MPM simulation is so important to your method, maybe spend a short paragraph explaining it to everyone in the main body of the paper.

---

> ### Author Response · Authors · 2025-11-21
>
> We highly appreciate your acknowledgment and encouragement toward our work. Your comments are highly valuable to us.
>
> For W1:
>
> Thank you for your valuable and professional suggestions. We will further refine the writing in the final version to make the method description more concise and precise.
>
> Regarding the question **“why not leverage gradients from a differentiable simulator to jointly optimize the upper-level physical laws,”** our consideration is as follows: the objective of upper-level optimization is to search for the **symbolic constitutive law**, which is **discrete and non-differentiable**. As a result, traditional gradient-based optimization techniques are not directly applicable in this context. To address this challenge, we introduce **LLMs as intelligent operators**, leveraging its rich constitutive prior to propose and revise symbolic constitutive law. Combined with an **evolutionary search paradigm**, this enables iterative optimization of symbolic constitutive laws.
>
> Regarding modeling upper-level constitutive discovery as a **continuous and differentiable** problem, we believe that this is a promising and important future direction. We believe that approaches such as **symbolic regression (e.g., DSR [1])** could provide a pathway forward, and we plan to explore this in our future work.
>
>
> *[1] Deep symbolic regression: Recovering mathematical expressions from data via risk-seeking policy gradients*
>
>
> For Q1 and Q2:
>
> Thank you for your valuable suggestions and careful feedback. We have revised the corresponding sections in the main paper, and the updated content has been **highlighted with a blue background** for clarity.

---

> > ### Comment · Reviewer_HDyU · 2025-11-25
> >
> > Thanks! I'm very happy with your changes and with the updates sections and additional information. I've raised my score and confidence. Good luck with the submission! Hope this gets accepted. This is very impressive work! :)

---

> > > ### Author Response · Authors · 2025-11-26
> > > **Acknowledgments**
> > >
> > > We sincerely appreciate the reviewer’s strong recognition of our work, as well as the valuable comments provided despite a busy schedule. Thank you!

---

### Author Response · Authors · 2025-12-02
**Rebuttal Summary**

We sincerely thank the Program Chairs, Senior Area Chairs, and Area Chairs for their coordination throughout the review process, as well as the reviewers for their thorough and constructive feedback.
To facilitate the tracking of revisions during the rebuttal phase, we summarize below the key concerns raised by the reviewers along with our corresponding responses.

### **Reviewer HDyU:**

* **Key Concerns:**

  The reviewer noted that the current presentation of the contributions is not well organized and suggested providing further explanations of the MPM simulation in the main body of the paper.

* **Our Responses:**

  We revised the manuscript to improve the organization and clarity of the stated contributions (Section 1), and provided additional explanation of the MPM simulation (Section 2.1). All modifications are highlighted with a blue background for clarity.

* **Reviewer Reaction:**

  The reviewer expressed strong appreciation for our work, and increased both the rating and confidence (Rating: **8 → 10**, Confidence: **3 → 4**, Presentation: **2 → 3**).

### **Reviewer aJkm:**

* **Key Concerns:**

  The reviewer requested a clearer explanation of how our method differs from [1] and [2] to better highlight our contributions. The reviewer also indicated that they would be willing to raise their score once this concern is addressed.

* **Our Responses:**

  We provide a capability comparison table that directly contrasts VisionLaw with prior works, accompanied by detailed textual explanations to clarify the distinctions between VisionLaw and prior works. To further substantiate these differences, we also provided additional experiments in the updated  **Section 4.3.4** and **Appendix B.5**, with new content highlighted using a red background.

* **Reviewer Reaction:**

  The reviewer has not yet participated in the discussion.

*[1] LLM and Simulation as Bilevel Optimizers: A New Paradigm to Advance Physical Scientific Discovery*

*[2] NeuMA: Neural Material Adaptor for Visual Grounding of Intrinsic Dynamics*

### **Reviewer 9Waz:**

* **Key Concerns:**

  The reviewer raised many questions regarding implementation details and seemed to hold a misunderstanding about our method. Specifically, the reviewer seemed to consider that our method merely selects and combines existing constitutive models (e.g., von Mises plasticity, Drucker–Prager plasticity, StVK elasticity).

* **Our Responses:**

  We try to do our best to address reviewer's concerns. Many of the requested implementation details were already explicitly described in the original manuscript. Therefore, while carefully answering these questions, we also referred the reviewer to the corresponding sections in the paper for more comprehensive explanations.

  To clarify the reviewer’s misunderstanding, we emphasized that the goal of the upper-level optimization is **not** to combine existing constitutive models, as such models are insufficient for capturing the full diversity of real-world dynamic behaviors. Instead, we leverage the LLM's extensive constitutive prior to **construct, reformulate, and discover new constitutive forms** that more effectively capture the underlying dynamics of visual observations.

* **Reviewer Reaction:**

  The reviewer has not yet participated in the discussion.

### **Reviewer cSP2:**

* **Key Concerns:**

  The reviewer requested an analysis and ablation study comparing the search efficiency with and without the decoupled strategy. The reviewer also pointed out limitations of the current approach and offered constructive suggestions for future improvement.

* **Our Responses:**

  Following the reviewer’s request, we quantified the search efficiency and conducted additional ablation studies on the decoupled strategy. These results are presented in Table 1 of our response to Reviewer cSP2 and in the updated **Appendix B.6**.

* **Reviewer Reaction:**

  The reviewer noted that they have no further concerns and increased the rating (**6 → 8**).

### **To AC:**

Across the four reviews, the main concerns were centered on clarity of presentation (Reviewer HDyU), differentiation from prior work (Reivewer aJkm), methodological clarification (Reviewer 9Waz), and additional experimental analyses (Reviewer cSP2). All of these have been systematically addressed in our revisions.

The discussion phase shows a clear positive trend: two reviewers express strong appreciation of our work and have already increased their scores; another indicated that they would consider changing their score once the concern was addressed (though they have not yet participated in the discussion); and the remaining concerns—which appear to stem from misunderstandings—have been explained and clarified in our responses.

Overall, the rebuttal provides the necessary clarifications, experiments, and analyses to address the reviewers' concerns.

Thank you once again for your dedication to the community.

Best regards,

Authors

---

### Meta-Review · Area_Chair_dvbw · 2026-01-18

**Summary:**

This paper proposes VisionLaw, a bilevel framework for inferring interpretable intrinsic dynamics from visual observations. An LLM-driven upper level evolves symbolic constitutive laws, while a vision-guided lower level evaluates candidates using differentiable MPM simulation and rendering. Reviewers agreed that the problem is important and that the combination of symbolic law discovery with visual evaluation is novel. The method shows strong results on both synthetic and real-world datasets, including generalization to novel scenarios.

**Reviewer Concerns:**

Concerns focused on presentation clarity, differentiation from prior work, methodological details of the upper-level evolution, and additional experimental analysis. Presentation issues raised by Reviewer HDyU were addressed through restructuring and added explanations, leading to an explicit score increase. Reviewer aJkm questioned novelty relative to SGA and NeuMA and indicated willingness to raise their score if distinctions were clarified. The authors responded with a capability comparison table, additional experiments, and expanded discussion, though the reviewer did not reengage.

Reviewer 9Waz raised detailed technical concerns about the correctness of constitutive modeling, search efficiency, and reproducibility. These were addressed with clarifications, ablation studies on the decoupled evolution strategy, and explicit implementation details, but the reviewer did not return to the discussion. Reviewer cSP2 requested quantitative analysis of the decoupled strategy and raised robustness questions. The authors added controlled ablations and acknowledged limitations, and this reviewer stated they had no further concerns and increased their score.

Overall, the rebuttal substantially improved clarity and added missing analyses. Remaining issues are largely scope assumptions that are acknowledged by the authors.

**Reviewer Scores:**

- Reviewer HDyU increased their score from 8 to 10 after the revisions.
- Reviewer cSP2 increased their score from 6 to 8 and stated they had no further concerns.
- Reviewer aJkm initially scored the paper 4 and indicated openness to raising the score if concerns were addressed, but did not reengage.
- Reviewer 9Waz scored the paper 2 and did not reengage after the rebuttal.

Two reviewers are clearly above threshold after rebuttal, with the remaining concerns either clarified or not followed up.

---

### Decision · Program_Chairs · 2026-01-26

Accept (Poster)